# JudgeLM : Fine-tuned Large Language Models are Scalable Judges

## Abstract

Evaluating Large Language Models (LLMs) in open-ended scenarios is challenging due to existing benchmarks and metrics can not measure them comprehensively. To address this problem, we propose to fine-tune **LLM**s as scalable **judge**s (JudgeLM) to evaluate LLMs efficiently and effectively in open-ended benchmarks. We first propose a comprehensive, large-scale, high-quality dataset containing task seeds, LLMs-generated answers, and GPT-4-generated judgments for fine-tuning high-performance judges, as well as a new benchmark for evaluating the judges. We train JudgeLM at different scales from 7B, 13B, to 33B parameters, and conduct a systematic analysis of its capabilities and behaviors. We then analyze the key biases in fine-tuning LLM as a judge and consider them as position bias, knowledge bias, and format bias. To address these issues, JudgeLM introduces a bag of techniques including swap augmentation, reference support, and reference drop, which clearly enhance the judge's performance. JudgeLM obtains the state-of-the-art judge performance on both the existing PandaLM benchmark and our proposed new benchmark. Our JudgeLM is efficient and the JudgeLM-7B only needs 3 minutes to judge 5K samples with 8 A100 GPUs. JudgeLM obtains high agreement with the teacher judge, achieving an agreement exceeding 90% that even surpasses human-to-human agreement[1]. JudgeLM also demonstrates extended capabilities in being judges of the single answer, multimodal models, multiple answers, and multi-turn chat.

## 1 Introduction

Recent advancements in large language models (LLMs) have fostered significant interest due to their remarkable performance in following instructions and their broad capabilities in dealing with open-ended scenarios. Based on the open-source LLMs, including OPT (Zhang et al., 2022), Flan-T5 (Chung et al., 2022), LLaMA (Touvron et al., 2023a), and Pythia (Biderman et al., 2023), researchers propose numerous methods to align these models with human preferences through instruction fine-tuning. These aligned LLMs demonstrate enhanced abilities in comprehending human instructions and generating more coherent responses. Nonetheless, existing benchmarks (Hendrycks et al., 2020; Liang et al., 2022) and traditional metrics (Lin, 2004; Papineni et al., 2002; Zhang et al., 2019; Sellam et al., 2020; Yuan et al., 2021) do not adequately estimate the capabilities of LLMs in open-ended scenarios. Therefore, a new benchmark method that could evaluate LLMs comprehensively in open-ended tasks is needed.

Concurrent works are making efforts to explore various methods for evaluating the performance of LLM. The arena-format (Zheng et al., 2023) methods leverage crowdsourced platforms to extract anonymous LLM competition results. While evaluations by humans are trustworthy, they are also time-consuming and financially demanding. Some approaches (Chiang et al., 2023) utilize GPT-4 as a judge. Nevertheless, these methods grapple with challenges of potential data exposure and volatile API model transitions, potentially compromising the judge's reproducibility. PandaLM (Wang et al., 2023) attempts to fine-tune open-source LLMs for evaluating answers. However, limitations stemming from the model's size, training data quality, and inherent LLM biases, undermine the effectiveness of such fine-tuned models in the role of a judge.

---

[1]As a reference, the max agreement among humans in MT-bench (Zheng et al., 2023) is 82%.

In this paper, we propose to evaluate LLMs through fine-tuned open-source **LLM**s, which serve as scalable **judge**s (JudgeLM) achieving satisfactory agreement with the teacher judge. Our methodology incorporates scalable judges as evaluators in open-ended tasks, coupled with a high-quality dataset conducive to both training and evaluating the judge models. Within our framework, we adapt open-source LLMs to serve as judges and analyze their scaling ability in relation to model size (ranging from 7B to 33B) and volume of training data (extending from 3.5K to 100K). Our curated dataset comprises 105K seed questions, LLM answer pairs, and judgments from the teacher judge, GPT-4, as shown in Fig. 1a. Note that we generated two judgments for each seed task with and without reference answers. This dataset is partitioned, with 100K seed questions allocated for training (×2 larger than PandaLM) and the remainder for validation (×29 larger than PandaLM).

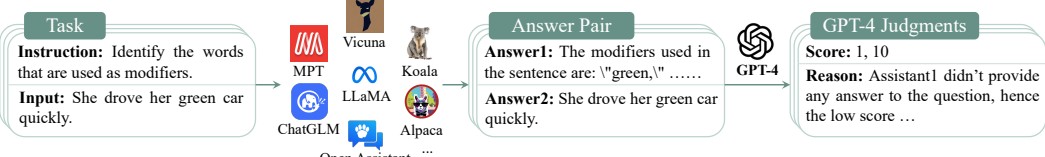

(a) Data generation pipeline of our JudgeLM. We first collect 105K seed tasks as questions. Then, we extract answers from 11 LLMs and randomly sample a pair of answers from the answer set. Last, we input the tasks, the sampled answer pairs, and optionally reference answers to GPT-4, which generates scores and detailed reasons as a judge teacher.

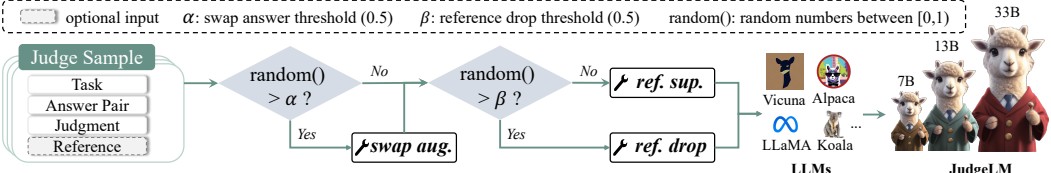

(b) An illustration of the JudgeLM's fine-tuning and various functions. We use generated judge samples to fine-tune LLMs as scalable judges. When fine-tuning LLMs as judges, we also propose swap augmentation, reference support, and reference drop to address the position bias, knowledge bias, and format bias, respectively.

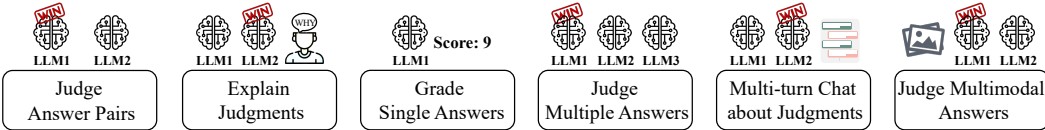

(c) An illustration of various functions of our JudgeLM.

Figure 1: An overview of our scalable JudgeLM including data generation, fine-tuning, and various functions.

Utilizing LLMs as judges inevitably introduces biases such as position bias (favoring answers in specific positions), knowledge bias (over-reliance on pre-trained knowledge), and format bias (optimal performance only under specific prompt formats) as shown in Fig. 8, 10, 12, 13. We propose methods to address them. Moreover, our JudgeLM system presents extended capabilities as shown in Fig. 1b, including grading single answers, judging multiple answers, judging multimodal models, and multi-turn chat.

In contrast to arena-format methodologies, our approach is rapid and has a low cost. For instance, a model like JudgeLM-7B requires only 8 A100 GPUs and can evaluate 5000 response pairs in just 3 minutes. In comparison to closed-source LLM judges, JudgeLM ensures reproducibility and protects user privacy. When compared to concurrent open-source LLM judges, our system explores both the scaling ability and biases in LLM fine-tuning. Furthermore, the dataset we introduce stands as the most diverse and high-quality one, significantly benefitting subsequent research in judge model investigations.

Our main contributions can be summarized as follows:

- We introduce a high-quality, large-scale dataset for judge models, enriched with diverse seed tasks, LLMs-generated answers, and detailed judgments from GPT-4, laying the foundation for future LLMs evaluating research.

- We propose JudgeLM, a scalable language model judge, designed for evaluating LLMs in open-ended scenarios. It achieves an agreement exceeding 90% that surpasses the human-to-human agreement. Our JudgeLM also has broad capacities to deal with extended tasks.

- We analyze the biases inherent to LLM judge fine-tuning and introduce a series of methods to address them. Our methods significantly improve the consistency of the model in different cases, making the JudgeLM more reliable and flexible.

## 2 Related Works

### 2.1 Instruction Fine-tuning of Large Language Models

With the development of large language models (LLMs), researchers find that fine-tuning pre-trained LLMs such as GPT-3 (Brown et al., 2020), T5 (Raffel et al., 2020), OPT (Zhang et al., 2022), and PaLM (Chowdhery et al., 2022) enables LLMs to follow human instructions and help with open-ended tasks. The instruction fine-tuned LLMs such as InstructGPT (Ouyang et al., 2022), ChatGPT (OpenAI, 2022), FLAN-T5 (Chung et al., 2022), FLAN-PaLM (Chung et al., 2022), OPT-IML (Iyer et al., 2022), and GPT-4 (OpenAI, 2023) exhibit stronger ability in zero-shot or few-shot tasks than their base models. After Meta released the powerful open-source LLM LLaMA (Touvron et al., 2023a) and LLaMA2 (Touvron et al., 2023b), lots of instruction fine-tuning works based on LLaMA or LLaMA2 were proposed in the natural language generation or multimodal generation domain, such as Alpaca, Vicuna (Chiang et al., 2023), OpenFlamingo (Awadalla et al., 2023), LLaMA-Adapter (Zhang et al., 2023), and Emu (Sun et al., 2023). Our JudgeLM also belongs to the LLaMA family and takes the Vicuna series as base models. Our JudgeLM follows the instruction fine-tuning manner to create LLM judges and proposes to model the judgment-generation task as "grading, judging, and reasoning". We further collect a high-quality, large-scale dataset for research in judging the performance of LLMs.

### 2.2 Evaluation of Large Language Models

As many open-source large language models (LLMs) and their fine-tuned variants are proposed and present remarkable performance on various tasks, evaluating the capabilities of LLMs becomes a popular and challenging task. To address this problem, Chatbot Arena (Zheng et al., 2023) aims to build a crowdsourced platform that ranks the LLMs through pairwise comparison and Elo rating. The crowdsourced way to evaluate LLMs has more reliable results but faces high costs and low efficiency. Vicuna (Chiang et al., 2023) uses GPT-4 as a judge to select the better answer. Although the GPT-4-based method can judge LLMs like a human expert, the API-based methods have potential risks of data leakage and unstable performance. Zeno Build (Alex & Graham, 2023) proposes to evaluate LLMs at a customer service dataset, but using traditional metrics such as ChrF (Popović, 2015) and BERTScore (Zhang et al., 2019) can not fully evaluate the answers of LLMs in open-ended tasks. Besides, PandaLM (Wang et al., 2023) developed a judge model based on LLaMA (Touvron et al., 2023a) to compare answers produced by LLMs. When serving as judges, PandaLM achieves an accuracy close to ChatGPT (OpenAI, 2022) but it only has a 7B model size that limits its performance further. Our JudgeLM contains scalable judges from 7B-parameter to 33B-parameter and achieves state-of-the-art performance in both PandaLM and our benchmarks. Furthermore, researchers can use the proposed JudgeLM locally which ensures reproducibility and data security.

## 3 Dataset

High-quality, large-scale datasets are crucial for effectively fine-tuning large language models (LLMs) to act as evaluative judges. However, the concurrent datasets, such as the one by Wang et al. (2023), present limitations in terms of diversity and the granularity of judgment criteria. To address this, we introduce a novel dataset replete with a rich variety of seed tasks, comprehensive answers from modern LLMs, answers' grades from the teacher judge, and detailed reasons for judgments. Section 3.1 elucidates the data generation process, while Section 3.2 delineates the methods adopted for training and evaluation using our dataset.

## 3.1 DATA GENERATION

---

**Input**

**Question**
If Rosslyn Mountain Boys had produced two more albums, how many albums would they have produced? Rosslyn Mountain Boys was a band that backed up a number of nationally known singers, and released two albums of their own.

**Answer 1**
4

**Answer 2**
Rosslyn Mountain Boys produced

> **Ground Truth**
> Rosslyn Mountain Boys produced two albums on their own, so if they produced two more albums then they would have produced four albums.

**Traditional Metrics**

| Answer | Rouge-1↑ | Rouge-2↑ | Rouge-L↑ | BLEU↑ | BERTScore↑ | BLEURT↑ | BARTScore↑ |
|--------|----------|----------|----------|-------|------------|---------|------------|
| 1 | 0.00 | 0.00 | 0.00 | 0.00 | 0.62 | -1.39 | -5.33 |
| 2 | 0.30 | 0.24 | 0.30 | 0.52 | 0.76 | -0.78 | -3.76 |

**Our Judgment Output**

**Answer 1's Score: 10**            **Answer 2's Score: 1**
Assistant 2's response is incomplete and does not provide any information or answer to the question, hence the low score. On the other hand, Assistant 1's response is accurate and directly answers the question. The Rosslyn Mountain Boys produced two albums, and if they had produced two more, they would have produced a total of four albums. Therefore, Assistant 1 receives a perfect score.

---

Figure 2: The input and output of the proposed JudgeLM data sample. By comparing the answers with ground truth, traditional metrics can not judge answers accurately. However, the LLM judges can understand the questions and answers and give accurate scores and reasons.

The primary objective of our data generation is to create a large-scale and diversified dataset that maximizes the evaluative capabilities of judge models. We sample 105K instruction seed tasks from a large-scale set that contains Alpaca-GPT4 (Peng et al., 2023), Dolly-15K (Conover et al., 2023), GPT4All-LAION (Anand et al., 2023), and ShareGPT. To enhance the heterogeneity of the dataset, answers are collated from 11 leading open-source LLMs including, but not limited to, LLaMA (Touvron et al., 2023a), Alpaca, and Vicuna (Chiang et al., 2023). Following this, we amalgamate LLM-generated answers with the reference answer to create answer sets. Pairs are randomly selected from the sets, upon which, fine-grained scores and detailed reasons are assigned by the advanced teacher model, GPT-4. To ensure robust and comprehensive judgments, we utilize detailed templates as demonstrated in Fig. 3. Additionally, to allow the model to judge with reference answers, the reference-inclusive template is employed as Fig. 4. This encourages the model to integrate external knowledge during the evaluative process.

## 3.2 TRAINING AND EVALUATING

To better utilize our dataset to train and evaluate the judge models, we partition it into a training split and a validation split. The training set contains 100K judge samples, while the validation set has 5K. We then introduce the way we use this dataset to train and evaluate, respectively.

**Training.** The training process of JudgeLM adheres to the instruction fine-tuning paradigm. As illustrated in Fig. 2, the model is fed a question alongside a pair of answers, and an optional reference

answer, yielding outputs comprising scores and detailed reasons. It is imperative to note the significance of a detailed crafted prompt template to harness the full potential of JudgeLM's instruction-following ability. Distinct input templates cater to scenarios with and without references, as depicted in Fig. 3 and Fig. 4 respectively.

To further analyze the scaling ability of our JudgeLM, we fine-tune JudgeLM with sizes of 7B, 13B, and 33B parameters. The specific hyperparameters are enumerated in Table 8. As for the scaling analysis for dataset size, we also fine-tune JudgeLM on varying data scales including 3.5K, 10K, 30K, and 100K samples. JudgeLM demonstrates scaling ability both in terms of parameter size and data volume.

**Evaluating.** For the judge's result, we model it as "grading, judging, and reasoning". The judge model first generates scores for answer pairs. Subsequently, we can get the judge result from three situations: "Answer 1 wins" if the answer 1's score is higher than the answer 2's, "Answer 2 wins" if the answer 2's score is higher, or "Tie" if the scores of two answers are the same. Last, the model generates detailed reasons if needed. The advantage of this modeling is that the judge model just needs little time to grade and judge, and generates time-consuming reasoning optionally.

For the metrics, we employ the objective metrics and reliability metrics to evaluate the judge models comprehensively. For the objective metrics, we compute the agreement, precision, recall, and F1-score between the model's judge results and those of the teacher. This provides insights into the alignment of judge models with established benchmarks, such as GPT-4 or human experts. As for reliability metrics, we first compare the results before and after swapping LLM answers. Then we calculate the consistency to measure the judge model's reliability. Last, we further calculate the metrics like "bias toward 1st", "bias toward 2nd", and "delta bias" to get insights from specific position biases and their variance.

## 4 INHERENT BIASES

In this paper, we also study the inherent biases that influence the reliability of fine-tuned LLM judges through reliability metrics and visualizations.

**Position Bias.** Position bias means that the LLM judges prefer answers in a certain position and it widely exists in natural language processing tasks (Ko et al., 2020; Wang et al., 2018) and decision-making of humans (Blunch, 1984; Raghubir & Valenzuela, 2006). The powerful LLMs, ChatGPT and GPT-4, also face this challenge when working as judges (Wang et al., 2023; Zheng et al., 2023). As the qualitative and quantitative results shown in Fig. 8 and Table 5, JudgeLM also faces the position bias and prefers the first answer when swapping the positions of answers.

**Knowledge Bias.** Knowledge bias arises when the pre-trained data lacks the knowledge of some seed tasks or induces possibly undesirable knowledge (Ko et al., 2020) that could degenerate the generative capabilities of LLMs. Fig. 10 provides an example that LLM judges can not give correct judgments to open-ended tasks if they lack related truth.

**Format Bias.** Researchers expect that the judge model can make judgments based on pre-trained knowledge when the reference is not available and can make judgments following the reference when it is available. However, our experiments revealed that judge models have a specific preference for the fine-tuning format whether with references or not. We name the situation that a judge fine-tuned without reference but validated with reference as a mismatched format, and vice versa. As shown in Fig. 12, Fig. 13, and Table 6 the judge models perform badly in mismatched formats. We hypothesize that the reason for format bias is that judge models are overfitting with the fixed fine-tuned template format.

## 5 MEHTODS

In evaluating LLM-generated answers for a seed question, the LLM judge aims to determine the superior answer from a pair of candidates. Motivated by recent methods (Touvron et al., 2023a; Chiang et al., 2023; Ouyang et al., 2022), we present JudgeLM, a scalable judge model, and address inherent biases in such models. Our methodology is depicted in Fig. 1b. The subsequent sections provide a detailed breakdown of our approach.

## 5.1 Swap Augmentation

MT-bench (Zheng et al., 2023) and PandaLM (Wang et al., 2023) alleviate the position bias by judging twice with original and reverse order. These methods regard the result as a tie if the judgments are not the same. This kind of method casting double time to evaluate, can be regarded as a compromise and does not fix the inherent position bias of LLMs.

Intuitively, swapping the positions at the fine-tuning stage could push the judge model to pay more attention to the contents of answers rather than positions. Leveraging our structured judge data, we can easily swap the positions of answers to generate a new input sample. Correspondingly, we also swap the scores and question indexes of the judgment from the teacher (i.e., GPT4) to get the new ground truth. As shown in Fig. 15, the augmented judge sample keeps the same results but exchanges the positions of answers. Overall, it is simple but effective to augment the training data and address position bias. The JudgeLM-with-swap-augmentation can give good judgment to the same judge sample as shown in Fig. 9.

## 5.2 Reference Support

Introducing external knowledge is an intuitive way to make up for the lack of related pre-trained knowledge. To do so, we propose the reference support method to teach the model to judge with the help of reference answers. We collect reference answers for all judge samples and re-generate reference-guided judgments by GPT-4. Please note that GPT-4 also gives different scores and judgments for most judge samples with or without references. This proves that the differences between pre-trained knowledge and reference answers greatly impact judgments. As shown in Fig. 11, the JudgeLM with reference support can avoid factual errors and give reliable judgments. Furthermore, introducing reference support to LLM judges can simply insert judge preferences. JudgeLM with reference support training can flexibly set reference answers with different preferences for different scenarios and needs. As shown in Fig. 16, changing reference answers does not need extra training and makes JudgeLM more flexible to different preferences.

## 5.3 Reference Drop

To address the format bias, we introduce a method, named reference drop, in which we randomly drop the training sample with reference and use the corresponding sample without reference. As shown in Fig. 14, judge models with reference drop can alleviate the overfitting for fine-tuning formats and give fair judgments with or without reference. Furthermore, the reference drop method also makes the judge model easy to use and decreases the cost of fitting into different formats.

## 6 Experiments

We study the performance of JudgeLM as follows: Section 6.1 presents the main results of JudgeLM comparing with concurrent methods, Section 6.2 analyzes the scaling ability of JudgeLM from both model sizes and data scales, and Section 6.3 shows ablation studies of proposed methods in detail.

## 6.1 Main Results

**Comparison on JudgeLM Benchmark.** We first evaluate the proposed JudgeLM on our $val$ set. As shown in Table 1, we give the quantitative results of GPT-3.5, PandaLM-7B, and our JudgeLM with three model sizes. Among them, GPT-3.5 is used in the form of APIs with the help of templates in Fig. 3 and Fig. 4. PandaLM-7B is deployed with the released checkpoint and template. These two methods could be regarded as zero-shot methods because they are not fine-tuned by the JudgeLM dataset. Our JudgeLMs are fine-tuned with proposed methods, i.e., swap augmentation, reference support, and reference drop. So, they can handle the situations with or without references simultaneously. It can be observed that our JudgeLM-7B outperforms PandaLM-7B in all metrics, and even surpasses GPT-3.5. Furthermore, the proposed JudgeLM-33B exhibits the most powerful judge ability.

**Comparison on PandaLM Benchmark.** We also zero-shot evaluate our JudgeLM on the PandaLM $test$ set. PandaLM uses the result of three human annotators' votes as ground truth. It requires

Table 1: Main results for our JudgeLM and concurrent methods on our *val* set, which uses GPT-4 annotation results as ground truth.

| Methods | Agreement ↑ (w/ GPT-4) | Precision ↑ (w/ GPT-4) | Recall ↑ (w/ GPT-4) | F1 ↑ (w/ GPT-4) | Consistency ↑ (w/ swap.) |
|---|---|---|---|---|---|
| *Judge w/o reference.* | | | | | |
| GPT-3.5 | 73.83 | 70.70 | 52.80 | 52.85 | 68.89 |
| PandaLM-7B | 68.61 | 40.75 | 38.82 | 39.41 | 74.78 |
| JudgeLM-7B | 81.11 | 69.67 | 78.39 | 72.21 | 83.57 |
| JudgeLM-13B | 84.33 | 73.69 | 80.51 | 76.17 | 85.01 |
| JudgeLM-33B | 89.03 | 80.97 | 84.76 | 82.64 | 91.36 |
| *Judge w/ reference.* | | | | | |
| GPT-3.5 | 71.46 | 56.86 | 51.12 | 51.14 | 62.94 |
| PandaLM-7B | 63.77 | 39.79 | 34.82 | 35.18 | 55.39 |
| JudgeLM-7B | 84.08 | 75.92 | 82.55 | 78.28 | 84.46 |
| JudgeLM-13B | 85.47 | 77.71 | 82.90 | 79.77 | 87.23 |
| JudgeLM-33B | 89.32 | 84.00 | 86.21 | 84.98 | 92.37 |

Table 2: JudgeLM zero-shot evaluation results on PandaLM *test* set, which uses human annotation results as ground truth. "*" means the results are reported in PandaLM (Wang et al., 2023)

| Methods | Accuracy ↑ | Precision ↑ | Recall ↑ | F1 ↑ |
|---|---|---|---|---|
| *zero-shot methods.* | | | | |
| GPT-3.5* | 62.96 | 61.95 | 63.59 | 58.20 |
| GPT-4* | 66.47 | 66.20 | 68.15 | 61.80 |
| *Fine-tuned on PandaLM train set.* | | | | |
| PandaLM-7B* | 59.26 | 57.28 | 59.23 | 54.56 |
| *Ours (zero-shot).* | | | | |
| JudgeLM-7B | 65.07 | 66.89 | 71.95 | 61.92 |
| JudgeLM-13B | 68.97 | 68.21 | 74.15 | 65.12 |
| JudgeLM-33B | 75.18 | 69.30 | 74.93 | 69.73 |

swapping the positions of two answers to perform inference twice and the conflicting evaluation results are modified to 'Tie'. Following the manner of this dataset, we present the zero-shot results of JudgeLM in Table 2. It can be observed that the JudgeLM-7B outperforms GPT-3.5 and PandaLM-7B. When compared with GPT-4, JudgeLM-7B has lower accuracy and higher Precision, Recall, and F1-score than GPT-4. Furthermore, JudgeLM-33B achieves higher results than GPT-4, which demonstrates that fine-tuned JudgeLM can outperform its teacher in this specific task.

**Efficiency comparison.** To further compare the efficiency between our JudgeLM and PandaLM, we conduct experiments on our *val* set to display the time cost using the same machine with 8 NVIDIA-A100 (40G) GPUs. As shown in Table 3, we display the methods and model sizes in the first and second columns. The third column shows the needed GPUs for each judge model. The models with 7B or 13B parameters run on 1 A100 GPU with 40G memory while the 33B-parameter needs 2 GPUs. The fourth column shows whether the methods can judge answers in parallel. PandaLM only runs on 1 GPU because it does not support parallel running but all our JudgeLM can run in parallel to fully use all the 8 GPUs. The fifth column indicates whether judge reasons are generated at runtime. Our JudgeLM can choose whether to generate reasons but PandaLM must generate reasons that make it too time-consuming. The sixth column presents the total time cost including the time of loading checkpoints, judging samples, and calculating metrics. It can be observed that our JudgeLM is much more flexible and efficient than PandaLM.

## 6.2 SCALING ANALYSIS OF JUDGELM

In this section, we analyze the scaling ability of the plain JudgeLM (without the proposed methods) on our *val* set without reference as illustrated in Table 4. As we increase the model size and data scale, we can observe the metrics increase. It demonstrates that the proposed JudgeLM is scalable

Table 3: Efficiency comparison for our JudgeLM and PandaLM on our *val* set. We use a machine with 8 Nvidia-A100 GPUs with 40G memory to evaluate their efficiency.

| Methods | model size | GPUs per model | parallel judge? | generate reason? | total time |
|---|---|---|---|---|---|
| PandaLM | 7B | 1 | ✗ | ✓ | 6 hrs 40 mins |
| ***Ours.*** | | | | | |
| JudgeLM | 7B | 1 | ✗ | ✓ | 6 hrs 40 mins |
| JudgeLM | 7B | 1 | ✗ | ✗ | 24 mins |
| JudgeLM | 7B | 1 | ✓ | ✓ | 50 mins |
| JudgeLM | 7B | 1 | ✓ | ✗ | 3 mins |
| JudgeLM | 13B | 1 | ✓ | ✗ | 5 mins |
| JudgeLM | 33B | 2 | ✓ | ✗ | 15 mins |

Table 4: Performance analysis for the scaling JudgeLM on our *val* set.

| Judge Size | Data Scale | Agreement ↑ (w/ GPT-4) | Consistency ↑ (w/ swap.) | Bias ↓ toward 1st | Bias ↓ toward 2nd | Delta bias ↓ |
|---|---|---|---|---|---|---|
| 7B | 3.5k | 75.87 | 73.45 | 19.83 | 6.72 | 13.11 |
| 7B | 10k | 78.89 | 78.25 | 17.3 | 4.45 | 12.85 |
| 7B | 30k | 81.43 | 80.89 | 14.54 | 4.57 | 9.97 |
| 7B | 100k | 83.71 | 82.62 | 12.31 | 5.07 | 7.24 |
| 13B | 3.5k | 80.61 | 78.91 | 14.68 | 6.41 | 8.27 |
| 13B | 10k | 83.19 | 81.9 | 13.42 | 4.68 | 8.74 |
| 13B | 30k | 84.39 | 82.99 | 11.96 | 5.05 | 6.91 |
| 13B | 100k | 85.87 | 83.01 | 11.53 | 5.46 | 6.07 |
| 33B | 3.5k | 85.38 | 85.16 | 9.34 | 5.5 | 3.84 |
| 33B | 10k | 87.49 | 86.4 | 8.32 | 5.28 | 3.04 |
| 33B | 30k | 88.84 | 87.34 | 7.57 | 5.09 | 2.48 |
| 33B | 100k | 90.06 | 87.93 | 6.85 | 5.22 | 1.63 |

and can reach up to 90.06% agreement and 87.93% consistency with 33B-parameter and 100K fine-tuning data.

## 6.3 ABLATION STUDY

In this section, we present the ablation studies of the proposed methods. For all ablation studies, we use JudgeLM-7B as the base model and 3.5K data for fine-tuning. Based on this baseline, we analyze the improvements brought by swap augmentation, reference support, and reference drop.

**Improvements of Swap augmentation.** As shown in Table 5, swap augmentation can bring comprehensive improvements to the baseline model. It improves consistency by 5.44%, which demonstrates that swap augmentation can reduce the influence of position bias and push the judge to pay more attention to the contents of answers.

**Improvements of Reference support.** As shown in the rows with the matching format of Table 6, JudgeLM fine-tuned with reference support exhibits superior performance on every metric. It demonstrates that the introduction of reference answers induces the judge to rely on external knowledge and addresses the limitation of pre-trained knowledge.

**Improvements of Reference drop.** As shown in Table 6, baselines can not reach satisfactory performance when facing mismatched formats. With the help of the reference drop, the JudgeLM can handle both the format with or without reference and achieve higher agreement and consistency. It demonstrates that reference drop can address the format bias and avoid the JudgeLM overfitting to a single format.

## 6.4 EXTENSIONS OF JUDGELM

Not only judging answer pairs, but our JudgeLM can also be applied to many other extended tasks, such as grading a single answer, judging and ranking multiple answers, evaluating multimodal models, and multi-turn chat about judgments.

Table 5: Ablation study for the swap augmentation on our *val* set.

| Methods | Agreement ↑ (w/ GPT-4) | Consistency ↑ (w/ swap.) | Bias ↓ toward 1st | Bias ↓ toward 2nd | Delta Bias ↓ |
|---|---|---|---|---|---|
| baseline | 75.87 | 73.45 | 19.83 | 6.72 | 13.11 |
| + swap aug. | 76.51 | 78.89 | 15.34 | 5.77 | 9.57 |

Table 6: Ablation study for the reference support and reference drop on our *val* set.

| Methods | *ft* w/ ref? | *val* w/ ref? | Agreement ↑ (w/ GPT-4) | Consistency ↑ (w/ swap.) | Bias ↓ toward 1st | Bias ↓ toward 2nd | Delta Bias ↓ |
|---|---|---|---|---|---|---|---|
| *matching format.* | | | | | | | |
| baseline | ✗ | ✗ | 75.87 | 73.45 | 19.83 | 6.72 | 13.11 |
| baseline | ✓ | ✓ | 80.15 | 81.23 | 11.55 | 7.22 | 4.33 |
| *mismatched format.* | | | | | | | |
| baseline | ✗ | ✓ | 73.09 | 67.75 | 29.44 | 2.81 | 26.63 |
| baseline | ✓ | ✗ | 75.69 | 73.40 | 20.89 | 5.71 | 15.18 |
| *w/ ref. drop.* | | | | | | | |
| baseline | ref. drop | ✗ | 76.86 | 77.13 | 17.30 | 5.57 | 11.73 |
| baseline | ref. drop | ✓ | 80.35 | 81.24 | 11.48 | 7.28 | 4.20 |

**Grading a single answer.** The Concurrent judge method (Wang et al., 2023) usually judges a pair of answers to decide which one is better or tie but they lack the ability to evaluate a single answer. Thanks to our judging mode of scoring first and then calculating the judging results, our JudgeLM provides an alternative practice to grade a single answer by slightly modifying the template as shown in Fig. 5. By putting the reference answer in the first position and giving it a full grade as a prior, JudgeLM can give quantitative fine-grained evaluations as shown in Fig. 17.

**Judging multiple answers.** To get the optimal ranking for N answers from different LLMs, other judge models need to call the model $O(n^2)$ times to get the full matrix, which is a much less efficient solution. We attempt to resolve this limitation by extending our JudgeLM to process multiple answers at the same time. We first need to modify the template as shown in Fig. 5. As shown in Fig. 18, JudgeLM can judge and rank the multiple answers within the context limit of LLM.

**Judging multimodal models.** Traditional multimodal evaluation needs prediction match the ground truth exactly. For some open-ended questions, a human-like evaluator is needed to determine whether the prediction is close to the ground truth range. Our JudgeLM provides good practice for multimodal evaluation by a slightly modified template as shown in Fig. 7. Thanks to its capacity to judge open-ended answers, our JudgeLM can also perform well in judging multimodal models, as shown in Fig. 21.

**Multi-turn chat about judgments.** It is worth noting that fine-tuning with judge samples does not compromise the multi-turn chat ability extended from base models. As illustrated in Fig. 19 and Fig. 20, our JudgeLM retains the capability to engage in meaningful dialogues with users, providing them with a richer context, detailed information, additional examples, and specific details.

## 7 CONCLUSION

In this paper, we propose JudgeLM as scalable judges for evaluating LLMs in open-ended tasks efficiently, achieving state-of-the-art judge performance on two benchmarks. The three key biases in fine-tuning LLMs as judges are analyzed and addressed by the proposed methods. We hope our work can motivate more studies to explore the judge models for LLMs in open-ended tasks and build more powerful LLMs with guidance from judge models.

**Limitations.** Although the proposed JudgeLM achieves encouraging performance and efficiency, the cost of the judge dataset limits further scaling up in the judge dataset. Currently, we spend about 4000 dollars to provide 100K high-quality GPT-4-generated judge data to the public. We expect to further improve the performance of judge models with the help of synthetic judge data.

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

# A APPENDIX

## A.1 MORE ABOUT DATASET

**Dataset Usage Scope** We emphasize that the JudgeLM dataset is intended only for academic research and any commercial use is prohibited. Because the OpenAI's terms prohibit developing models that compete with OpenAI, the instruction-tuning datasets generated by the OpenAI's API, i.e., Alpaca, PandaLM, etc., all follow this rule.

**Question Type of Validation Set** We count the distribution of questions in the JudgeLM *val* set as shown in Table 7.

Table 7: Distribution of questions in JudgeLM *val* set

|  | count | percentage |  | count | percentage |
|---|---|---|---|---|---|
| culture | 233 | 4.66% | planning | 309 | 6.18% |
| recommendation | 482 | 9.64% | roleplay | 77 | 1.54% |
| finance | 142 | 2.84% | coding | 201 | 4.02% |
| science | 393 | 7.86% | health | 278 | 5.56% |
| technique | 42 | 0.84% | writing | 625 | 12.50% |
| common-sense | 373 | 7.46% | hardware | 130 | 2.60% |
| art | 335 | 6.70% | history | 243 | 4.86% |
| math | 250 | 5.00% | geography | 199 | 3.98% |
| private-matter | 421 | 8.42% | others | 63 | 1.26% |
| law | 204 | 4.08% | total | 5000 | 100.00% |

## A.2 FINE-TUNING SETTINGS

We list the hyper-parameters we used, as shown in Table 8.

Table 8: JudgeLM fine-tuning setting.

| config | JudgeLM / -7B / -13B / -33B |
|---|---|
| base model | Vicuna / -7B / -13B / -33B |
| model max length | 2048 |
| fine-tuning data source | JudgeLM-100K |
| learning rate | 2e-5 |
| learning rate schedule | cosine decay |
| optimizer | AdamW Kingma & Ba (2014); Loshchilov & Hutter (2019) |
| optimizer hyper-parameters | $\beta_1, \beta_2, \epsilon$ = 0.9, 0.999, 1e-8 |
| weight decay | 0.0 |
| GPU nums | 8 / 8 / 16 |
| batch size | 128 |
| training epochs | 3 |
| warmup ratio | 0.003 |
| numerical precision | bf16, tf32 |
| ZeRO optimizer Ramesh et al. (2021) | stage 3 |
| gradient checkpointing | True |

## A.3 ADDITIONAL EXPERIMENTS

**Judging Multimodal Models** To validate the generalization ability of JudgeLM on judging multimodal models. We also conduct experiments on a multimodal benchmark, i.e., MM-Vet (Yu et al., 2023), the quantitative results are shown in Table 9, and the qualitative results are shown in Fig. 22, Fig. 23, and Fig. 24. Please note that MM-Vet is a vision-language benchmark for evaluating large multimodal models, which uses GPT-4 or GPT-3.5 as a judge. The API-based judge takes the question text, ground-truth text, and the model's prediction as input, and makes judgments based on them. We first use GPT-4, GPT-3.5, and JudgeLM to judge the LLaVA's output (Liu et al., 2023), respectively. Then, we collect judgments from human annotators, whose judgments include three situations: completely correct, semi-correct, and completely wrong. Last, we compute the metrics between the LLM judges' judgments and human judgments, as shown in Table 9. It can be observed that JudgeLM outperforms GPT-4 (0-shot) and GPT-3.5 (7-shot). Besides, JudgeLM achieves 2.5% higher precision than GPT-4 (7-shot). Furthermore, JudgeLM can use large multimodal models, e.g., LLaVA (Liu et al., 2023), as the backbone for better processing the multimodal judging. We leave it as a future work.

Table 9: JudgeLM zero-shot evaluation results on MM-Vet.

| Methods | Accuracy | Precision | Recall | F1-score |
|---|---|---|---|---|
| GPT-4 (7-shot) | 95.58 | 88.63 | 87.79 | 88.04 |
| GPT-4 (0-shot) | 86.70 | 79.75 | 86.41 | 81.81 |
| GPT-3.5 (7-shot) | 83.03 | 76.14 | 74.84 | 73.62 |
| JudgeLM-33B (0-shot) | 91.74 | 91.08 | 85.58 | 87.26 |

**Judging Multiple Answers** As shown in Table 10, we calculate the consistency results in the judging form of answer pairs and multiple answers. We first generate answers on JudgeLM val set through 3 LLMs, i.e., Vicuna-13B, LLaMA-7B, and alpaca-7B, for evaluation. Then we use pairwise judging and multiple judging to grade answers and rank them, respectively. Last, we compute the consistency between the two ranking results. Please note that 'Error Rate@2' indicates the position orderings of two answers are different between the result of paired judgment and the result of multiple judgment, and 'Error Rate@3' means the position orderings of three answers are different.

Table 10: Performance of JudgeLM-33B in judging multiple answers on JudgeLM *val* set. We calculate the consistency between the pairwise judging results and multiple judging ones.

| | Consistency | Error Rate@2 | Error Rate@3 |
|---|---|---|---|
| JudgeLM-33B | 93.48 | 6.38 | 0.14 |

**More Efficiency Comparison** We further add an ablation study for JudgeLM without parallel judging to achieve more comprehensive ablations, as shown in Table 3. When generating reasons, JudgeLM-7B's efficiency is similar to PandaLM-7B. Furthermore, it can be observed that JudgeLM-7B is 16.65 times faster than PandaLM-7B when it does not generate reasons. The root cause of efficiency improvements is our modeling method, i.e., "grading, judging, and reasoning", and support for parallelization is an engineering optimization. The reason for the parallel judging is that we are committed to making JudgeLM an efficient tool for researchers and developers, benefiting the entire LLM community.

Table 11: Performance of JudgeLM-7B with explanation-first (CoT) or score-first (Ours) on JudgeLM $val$ set.

| Methods | Agreement ↑ (w/ GPT-4) | Consistency ↑ (w/ swap.) | Bias ↓ toward 1st | Bias ↓ toward 2nd | Delta Bias ↓ |
|---|---|---|---|---|---|
| score-first (Our) | 75.87 | 73.45 | 19.83 | 6.72 | 13.11 |
| explanation-first (CoT) | 75.54 | 74.39 | 15.05 | 10.56 | 4.50 |

**Ablation of Judging Form** We further evaluate the performance of JudgeLM-7B with explanation-first (CoT (Wei et al., 2022)) or score-first (Ours) in Table 11. JudgeLM with CoT performs similar agreement with our score-first baseline but with higher consistency, which means that explanation-first form, i.e., CoT, can alleviate the position bias of fine-tuned judges, but not bring significant agreement improvement. So, we choose the score-first method, which has slightly less consistency but more flexible usage.

**Biases Analysis on PandaLM** We further analyze the three biases on PandaLM and apply the proposed methods to it, as shown in Table 12 and Table 13. We first transfer the 3.5K JudgeLM train samples into PandaLM format. Then, we train the PandaLM baseline, i.e., PandaLM*, with LLaMA-7B and the 3.5K train samples in PandaLM format. It can be observed that the PandaLM* also has position bias, which prefers the last answer. Besides, PandaLM* also faces knowledge bias and format bias, as shown in Table 13. Last, we apply the proposed methods to PandaLM*. As shown in Table 12 and Table 13, the proposed methods can also improve the performance of PandaLM*.

Table 12: Ablation study of PandaLM for the swap augmentation on our $val$ set.

| Methods | Agreement ↑ (w/ GPT-4) | Consistency ↑ (w/ swap.) | Bias ↓ toward 1st | Bias ↓ toward 2nd | Delta Bias ↓ |
|---|---|---|---|---|---|
| PandaLM* | 71.30 | 69.59 | 9.43 | 20.98 | 11.55 |
| + swap aug. | 72.41 | 73.50 | 8.71 | 17.79 | 9.08 |

Table 13: Ablation study of PandaLM for the reference support and reference drop on our $val$ set.

| Methods | $ft$ w/ ref? | $val$ w/ ref? | Agreement ↑ (w/ GPT-4) | Consistency ↑ (w/ swap.) | Bias ↓ toward 1st | Bias ↓ toward 2nd | Delta Bias ↓ |
|---|---|---|---|---|---|---|---|
| *matching format.* | | | | | | | |
| PandaLM* | ✗ | ✗ | 71.32 | 69.59 | 9.43 | 20.98 | 11.55 |
| PandaLM* | ✓ | ✓ | 75.15 | 74.08 | 8.62 | 17.30 | 8.68 |
| *mismatched format.* | | | | | | | |
| PandaLM* | ✗ | ✓ | 68.67 | 65.00 | 15.82 | 19.18 | 3.36 |
| PandaLM* | ✓ | ✗ | 70.04 | 70.56 | 9.71 | 19.73 | 10.02 |
| *w/ ref. drop.* | | | | | | | |
| PandaLM* | ref. drop | ✗ | 71.88 | 71.56 | 11.24 | 17.20 | 5.96 |
| PandaLM* | ref. drop | ✓ | 75.93 | 74.77 | 9.22 | 16.01 | 6.79 |

**Comparison with GPT-4 Teacher** As shown in Table 14, we further list the metrics except for the agreement with GPT-4 itself. JudgeLM-33B achieves higher consistency than GPT-4, which demonstrates that JudgeLM-33B is more robust with position bias.

Table 14: Comparison between GPT-4 teacher and JudgeLM-33B on JudgeLM *val* set.

| Methods | Agreement ↑ (w/ GPT-4) | Consistency ↑ (w/ swap.) | Bias ↓ toward 1st | Bias ↓ toward 2nd | Delta Bias ↓ |
|---|---|---|---|---|---|
| GPT-4 | – | 85.82 | 6.10 | 8.10 | 2.00 |
| JudgeLM-33B | 89.03 | 91.36 | 5.55 | 3.09 | 2.46 |

Table 15: Performance of JudgeLM-33B with injected paragraphs as references on JudgeLM *val* set.

| Reference Paragraph | Agreement ↑ (w/ GPT-4) | Consistency ↑ (w/ swap.) | Bias ↓ toward 1st | Bias ↓ toward 2nd | Delta Bias ↓ |
|---|---|---|---|---|---|
| No | 89.32 | 92.37 | 3.62 | 4.01 | 0.39 |
| 50 words | 87.78 | 92.35 | 3.00 | 4.65 | 1.65 |
| 100 words | 87.69 | 91.84 | 2.62 | 5.54 | 2.92 |
| 200 words | 86.77 | 90.48 | 2.70 | 6.82 | 4.12 |
| 300 words | 86.25 | 89.26 | 3.13 | 7.61 | 4.48 |
| 400 words | 85.59 | 89.00 | 2.98 | 8.02 | 5.04 |

**Judging with Injected Paragraph** As shown in Table 15, we further inject original reference answers into paragraphs with different words, and evaluate JudgeLM-33B with the injected paragraphs as references in a zero-shot setting. It shows that JudgeLM-33B can use references in the retrieved form with a certain loss of agreement and consistency.

Table 16: JudgeLM fine-grained evaluation results on JudgeLM *val* set.

| | Accuracy | Precision | Recall | F1-score |
|---|---|---|---|---|
| factuality | 89.69 | 77.61 | 84.79 | 80.33 |
| fluency | 88.35 | 79.86 | 82.35 | 81.01 |
| novelty | 87.98 | 78.63 | 84.85 | 81.12 |
| helpfulness | 88.16 | 79.30 | 83.88 | 81.26 |
| all 4 aspects above | 87.92 | 78.70 | 81.78 | 80.08 |

**Judging Fine-grained Aspects** We further ask the JudgeLM to output the fine-grained results through the modified template. We also evaluate the JudgeLM-33B on the JudgeLM *val* set as shown in Table 16.

## A.4 MORE DISCUSSION

**Why format bias does not affect multiple judging?** As shown in Table 10, it can be seen that the judging of multiple answers does not receive a significant performance drop. We hold the viewpoint that judging multiple answers is an easy extension for JudgeLM, which does not change the basis for judging. As mentioned in '4 Inherent Biases - Format Bias', format bias means the model judging basis changes from pre-trained knowledge to reference, or vice versa. So, judging in mismatched situations faces format bias but judging multiple answers does not.

**How about evaluation on non-GPT-4 annotated benchmarks?** For a fair comparison, we also evaluate JudgeLM on the PandaLM test set in a zero-shot setting. The PandaLM test set is annotated by humans. As shown in Table 2, the zero-shot results of JudgeLM also outperform other judging methods, i.e., PandaLM, GPT-3.5, and GPT-4. Furthermore, JudgeLM also achieves a superior 0-shot judging performance on the multimodal benchmark with human annotation, i.e., MM-Vet, as shown in Table 9.

**How can we trust the evaluation results of LLM judges performing badly at certain reasoning tasks?** Nowadays, NLP researchers are still struggling with proposing LLMs with superior reasoning abilities. JudgeLM also needs the proposed reference sup method to enhance the judging ability for out-of-domain or counterfactual tasks, as shown in Fig. 11 and Fig. 16. Notably, the proposed JudgeLM can benefit from stronger foundation LLMs, e.g., the LLaMA2-7B-Chat-based Touvron

Table 17: Comparison of different base models for JudgeLM-7B on JudgeLM *val* set.

| Methods | Agreement ↑ (w/ GPT-4) | Precision ↑ (w/ GPT-4) | Recall ↑ (w/ GPT-4) | F1 ↑ (w/ GPT-4) | Consistency ↑ (w/ swap.) |
|---|---|---|---|---|---|
| ***Judge w/o reference.*** | | | | | |
| JudgeLM-7B w/ Vicuna | 81.11 | 69.67 | 78.39 | 72.21 | 83.57 |
| JudgeLM-7B w/ LLaMA2-chat | 83.87 | 73.43 | 80.06 | 75.91 | 85.17 |
| ***Judge w/ reference.*** | | | | | |
| JudgeLM-7B w/ Vicuna | 84.08 | 75.92 | 82.55 | 78.28 | 84.46 |
| JudgeLM-7B w/ LLaMA2-chat | 86.60 | 79.47 | 83.11 | 81.02 | 87.74 |

et al. (2023b) JudgeLM outperforms the original JudgeLM-7B on all metrics, as shown in Table 17. The research of judge models is critical for the development of LLMs and can benefit from advanced LLMs, establishing a positive cycle.

**Is Reference Drop a Hyper-parameter or an Independent Method?** We think the Reference Drop should be an independent method. At first, we argue that judging with or without references are two sub-benchmarks, which require judges to make judgments with internal knowledge or by comparing LLM-generated answers with a reference answer, respectively. The reference drop is not only a simple but effective hyper-parameter, but also an important method that bridges the two sub-benchmarks, which enables the JudgeLM to make judgments in different situations.

**What are the differences between PandaLM and JudgeLM?** Compared with PandaLM, our method has these different contributions:

- We introduce a high-quality, large-scale dataset for judge models, enriched with diverse seed tasks, LLMs-generated answers, and detailed judgments from GPT-4, laying the foundation for future LLMs evaluating research.

- We analyze the biases inherent to LLM judge fine-tuning and introduce a series of methods to address them. Our methods significantly improve the consistency of the model in different cases, making the JudgeLM more reliable and flexible.

- We analyze the scaling ability of the language model judge and propose the JudgeLM series, i.e., JudgeLM-7B, JudgeLM-13B, and JudgeLM-33B, for evaluating LLMs in open-ended scenarios.

- The proposed judging pattern, i.e., grading, judging, and reasoning, makes judging efficient, which only needs little time to grade and judge, and generates time-consuming reasons optionally.

- The proposed JudgeLM presents extended capabilities as shown in Fig. 1b, including grading single answers, judging multiple answers, judging multimodal models, and multi-turn chat.

### A.5 PROMPT TEMPLATES

We list all the prompt templates we used.

### A.6 CASE STUDIES

We list several case studies.

You are a helpful and precise assistant for checking the quality of the answer.
[Question]
{question}

[The Start of Assistant 1's Answer]
{answer_1}
[The End of Assistant 1's Answer]

[The Start of Assistant 2's Answer]
{answer_2}
[The End of Assistant 2's Answer]

[System]
We would like to request your feedback on the performance of two AI assistants in response to the user question displayed above.
Please rate the helpfulness, relevance, accuracy, level of details of their responses. Each assistant receives an overall score on a scale of 1 to 10, where a higher score indicates better overall performance.
Please first output a single line containing only two values indicating the scores for Assistant 1 and 2, respectively. The two scores are separated by a space. In the subsequent line, please provide a comprehensive explanation of your evaluation, avoiding any potential bias and ensuring that the order in which the responses were presented does not affect your judgment.

Figure 3: The template for judging answers without the reference.

You are a helpful and precise assistant for checking the quality of the answer.
[Question]
{question}

[Reference Answer]
{reference}

[The Start of Assistant 1's Answer]
{answer_1}
[The End of Assistant 1's Answer]

[The Start of Assistant 2's Answer]
{answer_2}
[The End of Assistant 2's Answer]

[System]
We would like to request your feedback on the performance of two AI assistants in response to the user question displayed above.
Based on the reference answer, please rate the helpfulness, relevance, accuracy, level of details of their responses. Each assistant receives an overall score on a scale of 1 to 10, where a higher score indicates better overall performance.
Please first output a single line containing only two values indicating the scores for Assistant 1 and 2, respectively. The two scores are separated by a space. In the subsequent line, please provide a comprehensive explanation of your evaluation, avoiding any potential bias and ensuring that the order in which the responses were presented does not affect your judgment.

Figure 4: The template for judging answers with the reference.

You are a helpful and precise assistant for checking the quality of the answer.
[Question]
{question}

[The Start of Assistant 1's Answer]
{reference}
[The End of Assistant 1's Answer]

[The Start of Assistant 2's Answer]
{single answer}
[The End of Assistant 2's Answer]

[System]
We would like to request your feedback on the performance of two AI assistants in response to the user question displayed above.
Please rate the helpfulness, relevance, accuracy, level of details of their responses. Each assistant receives an overall score on a scale of 1 to 10, where a higher score indicates better overall performance.
Please first output a single line containing only two values indicating the scores for Assistant 1 and 2, respectively. The two scores are separated by a space. In the subsequent line, please provide a comprehensive explanation of your evaluation, avoiding any potential bias and ensuring that the order in which the responses were presented does not affect your judgment.

[Response]
10

Figure 5: The template for grading a single answer. We set the reference answer in the position of Answer 1. Then, we set the score of the reference answer to 10. Last, the JudgeLM outputs the score of the single answer with such a prior.

You are a helpful and precise assistant for checking the quality of the answer.
[Question]
{question}

[The Start of Assistant 1's Answer]
{answer_1}
[The End of Assistant 1's Answer]

[The Start of Assistant 2's Answer]
{answer_2}
[The End of Assistant 2's Answer]

[The Start of Assistant 3's Answer]
{answer_3}
[The End of Assistant 3's Answer]

[System]
We would like to request your feedback on the performance of three AI assistants in response to the user question displayed above.
Please rate the helpfulness, relevance, accuracy, level of details of their responses. Each assistant receives an overall score on a scale of 1 to 10, where a higher score indicates better overall performance.
Please first output a single line containing only three values indicating the scores for Assistant 1, 2 and 3, respectively. The three scores are separated by a space. In the subsequent line, please provide a comprehensive explanation of your evaluation, avoiding any potential bias and ensuring that the order in which the responses were presented does not affect your judgment.

Figure 6: The template for judging multiple answers.

Please evaluate the following visual-based question-answer pair:

[Question]
{question}

[Reference Answer]
{reference}

[The Start of Assistant 1's Answer]
{answer_1}
[The End of Assistant 1's Answer]

[The Start of Assistant 2's Answer]
{answer_2}
[The End of Assistant 2's Answer]

[System]
Your task is to compare the predicted answer with the correct answer and determine if they match meaningfully. To accomplish the task, you must :
1. Focus on the meaningful match between the predicted answer and the correct answer.
2. Consider synonyms or paraphrases as valid matches.
3. Evaluate the correctness of the prediction compared to the answer.
4. If there are multiple correct answers, the predicted answer is considered correct as long as it is close to any of the answers

Figure 7: The template for multimodal judging.

---

**Input**

**Question**
Categorize a list of flowers.
Lilac, Daisy, Marigold, Chrysanthemum

**Answer 1**
Lilac: Shrub
Daisy: Herbaceous Perennial
Marigold: Annual
Chrysanthemum: Perennial

**Answer 2**
Flower:
- Lilac
- Daisy
- Marigold
- Chrysanthemum

**Our Judgement Output (where answer 1 is placed in the first position)**

**Answer 1's Score: 8          Answer 2's Score: 6**
Assistant 1 provided a more detailed and accurate response, correctly categorizing the flowers as shrubs, herbaceous perennials, annuals, and perennials. The assistant also provided a brief explanation of what these categories mean, which adds to the helpfulness and relevance of the response. However, the assistant could have provided more information about the characteristics of these flowers, such as their size, growth habit, and blooming time. On the other hand, Assistant 2 did not provide any information about the flowers' categories. The assistant simply listed the flowers without any additional information or context. While this response is technically correct, it lacks the detail and context that would make it more helpful and informative.

**Our Judgement Output (where answer 2 is placed in the first position)**

**Answer 2's Score: 8          Answer 1's Score: 6**
Assistant 2's response is concise and accurate, correctly categorizing the flowers as Lilac, Daisy, Marigold, and Chrysanthemum. However, the assistant could have provided more detail or explanation about the characteristics of each flower, which would have made the response more helpful.Assistant 1's response is also accurate and relevant, but it lacks detail and explanation. The assistant categorizes the flowers as Lilac (Shrub), Daisy (Herbaceous Perennial), Marigold (Annual), and Chrysanthemum (Perennial). However, these categories are not commonly used in the horticultural industry, and the assistant does not provide any explanation or context for these categories. This makes the response less helpful and informative.

Figure 8: Bad judgment caused by position bias. The answer placed in the first position always gets a higher score. The judge models generate reasons as possible from the perspective of making the scores reasonable.

**Input**

**Question**
Categorize a list of flowers.
Lilac, Daisy, Marigold, Chrysanthemum

**Answer 1**
Lilac: Shrub
Daisy: Herbaceous Perennial
Marigold: Annual
Chrysanthemum: Perennial

**Answer 2**
Flower:
- Lilac
- Daisy
- Marigold
- Chrysanthemum

**Our Judgement Output (where answer 1 is placed in the first position)**

**Answer 1's Score: 8          Answer 2's Score: 2**
Assistant 1's response is more helpful and accurate as it provides a categorization for each flower listed in the question. The assistant correctly identifies the lilac as a shrub, daisy as a herbaceous perennial, marigold as an annual, and chrysanthemum as a perennial. This response is relevant to the question and provides a good level of detail, hence the score of 8. On the other hand, ==Assistant 2's response is not helpful or accurate. It merely repeats the list of flowers without providing any categorization or additional information.== The response is not relevant to the question and lacks detail, hence the low score of 2.

**Our Judgement Output (where answer 2 is placed in the first position)**

**Answer 2's Score: 2          Answer 1's Score: 8**
Assistant 1's response is more helpful and relevant to the question. The user asked to categorize a list of flowers, and Assistant 2 correctly categorized each flower by its life cycle (annual, perennial, etc.). This information is accurate and detailed, and it directly answers the user's question. Therefore, Assistant 2 receives a score of 8. On the other hand, ==Assistant 2's response is not as helpful or relevant. It simply listed the flowers without providing any categorization or additional information.== This response does not answer the user's question and lacks detail, so Assistant 1 receives a score of 2.

Figure 9: Good judgment generated by the judge fine-tuned with swap augmentation. The judge can give judgments based on the content of answers rather than a certain position. The reason is convincing and reasonable.

**Input**

**Question**
Determine which companies are primarily cloud-software companies and which are semiconductor companies: Intel, Google, Workday, Zscaler, AMD, NVIDIA, and Texas Instruments.

**Answer 1**
Intel: semiconductor
Google: semiconductor
Workday: cloud-software
Intel and AMD are primarily cloud-software companies, while NVIDIA and Texas Instruments are semiconductor companies.
AMD: semiconductor
NVIDIA: semiconductor
Texas Instruments: semiconductor

**Answer 2**
Intel and AMD are primarily cloud-software companies, while NVIDIA and Texas Instruments are semiconductor companies.

**Our Judgement Output**

**Answer 1's Score: 2**       **Answer 2's Score: 8**
Assistant 1's answer is incorrect and confusing. Intel and AMD are not primarily cloud-software companies, and NVIDIA and Texas Instruments are not semiconductor companies. The information provided is not accurate or relevant to the question asked. On the other hand, Assistant 2's answer is more accurate and relevant. Intel and AMD are indeed primarily cloud-software companies, and NVIDIA and Texas Instruments are semiconductor companies. However, the answer could have been more detailed, explaining why these companies are classified as such.

Figure 10: Bad judgment caused by knowledge bias. This seed task is out of the judge model's pre-trained knowledge, so it can not be correctly judged by the judge model. The judge model gives contradictory reasons in the judgment.

**Input**

**Question**
Determine which companies are primarily cloud-software companies and which are semiconductor companies: Intel, Google, Workday, Zscaler, AMD, NVIDIA, and Texas Instruments.

**Reference Answer**
Intel, AMD, NVIDIA, and Texas Instruments are primarily semiconductor companies. Workday and Zscaler are fully cloud-software companies. Google is primarily a software company, however, they also develop custom-developed application-specific integrated circuits which are semiconductors."

**Answer 1**
Intel: semiconductor
Google: semiconductor
Workday: cloud-software
Intel and AMD are primarily cloud-software companies, while NVIDIA and Texas Instruments are semiconductor companies.
AMD: semiconductor
NVIDIA: semiconductor
Texas Instruments: semiconductor

**Answer 2**
Intel and AMD are primarily cloud-software companies, while NVIDIA and Texas Instruments are semiconductor companies.

**Our Judgement Output**

**Answer 1's Score: 8**          **Answer 2's Score: 3**
Assistant 2's answer is incorrect and misleading. They stated that Intel and AMD are primarily cloud-software companies, which is not true. Both Intel and AMD are semiconductor companies. Therefore, the accuracy, relevance, and helpfulness of Assistant 2's answer are low, resulting in a score of 3.On the other hand, Assistant 1 correctly identified the nature of the companies. They correctly identified Intel, AMD, NVIDIA, and Texas Instruments as semiconductor companies and Workday and Zscaler as cloud-software companies. However, they incorrectly identified Google as a semiconductor company, while Google is primarily a software company that also develops custom-developed application-specific integrated circuits which are semiconductors. Despite this minor error, Assistant 1's answer is much more accurate, relevant, and helpful than Assistant 2's, earning it a score of 8.

Figure 11: Good judgment generated by the judge model fine-tuned with reference support. Even though the judge model itself lacks related information, it can also give a reasonable judgment with the reference answer.

---

**Input**

**Question**
Generate a descriptive sentence to describe the painting
Monalisa

**Answer 1**
The Italian artist, Leonardo da Vinci created the most famous painting of all time. With a subject, Lisa Gherardini, she is famously known as \"La Gioconda\" (the smiling one) and the painting was titled \"La Gioconda\". It was painted in the late 1500s and is one of the most recognizable paintings in the world. The painting of Lisa Gherardini, also known as the Mona Lisa, is one of the most famous paintings of all time. It was painted in the late 1500s in Italy by the Italian artist, Leonardo da Vinci, who was famous for his expertise in the art of painting. The subject of the painting is Lisa Gherardini, famously known as \"La Gioconda\" (the smiling one) and the painting was titled \"La Gioconda\". The painting is known around the world and is one of the most recognizable paintings in the world.Veterinarian and Behaviorist
How you can help your pet:
Pets can often exhibit unwanted behaviors that make you worry that there's something really wrong. But the fact is, most problems stem from misunderstood communication. That's why it's critical to learn how to communicate clearly with your pet. I

**Answer 2**
The painting Monalisa was made by Leonardo da Vinci.

**Our Good Judgment (JudgeLM trained without reference)**

**Answer 1's Score: 4          Answer 2's Score: 7**
Assistant 1's response started off well, providing a detailed description of the painting "Monalisa" by Leonardo da Vinci. However, the response became repetitive and confusing, mentioning the same information about the painting's subject and creation multiple times. Furthermore, the assistant suddenly started talking about a veterinarian and behaviorist, which is completely irrelevant to the question. Therefore, the score is 4. ==Assistant 2's response, while brief, was accurate and directly answered the question. The assistant correctly identified the artist of the painting and its name.== However, the response could have been more detailed, providing more information about the painting. Therefore, the score is 7.

**Our Bad Judgment (JudgeLM trained with reference)**

**Answer 1's Score: 2          Answer 2's Score: 1**
Assistant 1's response is confusing and inaccurate. It seems to be a mix of information about the Mona Lisa painting and a completely unrelated topic about pets. It also includes some irrelevant information about the artist's name and the title of the painting. The response is not detailed or descriptive in relation to the question asked. Therefore, it gets a score of 2. ==Assistant 2's response is even less helpful. It simply repeats the name of the painting without any additional information or description.== It does not answer the question at all, hence it gets a score of 1.

Figure 12: Bad judgment caused by format bias. For judging without reference, the judge model trained without reference is matched, so it performs well. However, the judge model trained with reference is mismatched, so it performs badly.

**Input**

**Question**
Generate a descriptive sentence to describe the painting
Monalisa

**Reference Answer**
The Mona Lisa painting is a half-length portrait, painted by Leonardo da Vinci, that depicts a seated woman with a slight smile, enigmatic expression, and intricate details in her attire, set against a hazy and dreamlike backdrop.

**Answer 1**
The Italian artist, Leonardo da Vinci created the most famous painting of all time. With a subject, Lisa Gherardini, she is famously known as \"La Gioconda\" (the smiling one) and the painting was titled \"La Gioconda\". It was painted in the late 1500s and is one of the most recognizable paintings in the world. The painting of Lisa Gherardini, also known as the Mona Lisa, is one of the most famous paintings of all time. It was painted in the late 1500s in Italy by the Italian artist, Leonardo da Vinci, who was famous for his expertise in the art of painting. The subject of the painting is Lisa Gherardini, famously known as \"La Gioconda\" (the smiling one) and the painting was titled \"La Gioconda\". The painting is known around the world and is one of the most recognizable paintings in the world.Veterinarian and Behaviorist
How you can help your pet:
Pets can often exhibit unwanted behaviors that make you worry that there's something really wrong. But the fact is, most problems stem from misunderstood communication. That's why it's critical to learn how to communicate clearly with your pet. I

**Answer 2**
The painting Monalisa was made by Leonardo da Vinci.

**Our Bad Judgment (JudgeLM trained without reference)**

**Answer 1's Score: 8          Answer 2's Score: 2**
Assistant 1 provided a detailed and accurate description of the Mona Lisa painting, including its creation by Leonardo da Vinci, its subject, and its title. The assistant also provided additional information about the painting's fame and the artist's expertise. However, the assistant's response was cut off at the end, which is why it didn't receive a perfect score.==Assistant 2, on the other hand, did not provide any information about the Mona Lisa painting.== The assistant simply repeated the name of the painting, which is not helpful or informative. Therefore, Assistant 2 receives a low score.

**Our Good Judgment (JudgeLM trained with reference)**

**Answer 1's Score: 5          Answer 2's Score: 7**
Assistant 1's response started off well, providing a detailed description of the Mona Lisa painting, including the painting artist's name, the subject of the painting, and its significance. However, ==Assistant 1's response became repetitive that mentioning the same information multiple times and suddenly shifted to a completely unrelated topic about pets==, which is irrelevant to the question. Therefore, I would rate Assistant 1 a 5.==Assistant 2's response was brief and to the point, correctly identifying Leonardo da Vinci as the artist who painted the Mona Lisa.== However, it lacked detail and did not provide any additional information about the painting itself. Despite its brevity, Assistant 2's response was more relevant and accurate than Assistant 1's, so I would rate it a 7.

Figure 13: Bad judgment caused by format bias. For judging with reference, the judge model trained with reference is matched, so it performs well. However, the judge model trained without reference is mismatched, so it performs badly.

**Input**

**Question**
Generate a descriptive sentence to describe the painting
Monalisa

**Reference Answer**
The Mona Lisa painting is a half-length portrait, painted by Leonardo da Vinci, that depicts a seated woman with a slight smile, enigmatic expression, and intricate details in her attire, set against a hazy and dreamlike backdrop.

**Answer 1**
The Italian artist, Leonardo da Vinci created the most famous painting of all time. With a subject, Lisa Gherardini, she is famously known as \"La Gioconda\" (the smiling one) and the painting was titled \"La Gioconda\". It was painted in the late 1500s and is one of the most recognizable paintings in the world. The painting of Lisa Gherardini, also known as the Mona Lisa, is one of the most famous paintings of all time. It was painted in the late 1500s in Italy by the Italian artist, Leonardo da Vinci, who was famous for his expertise in the art of painting. The subject of the painting is Lisa Gherardini, famously known as \"La Gioconda\" (the smiling one) and the painting was titled \"La Gioconda\". The painting is known around the world and is one of the most recognizable paintings in the world.Veterinarian and Behaviorist
How you can help your pet:
Pets can often exhibit unwanted behaviors that make you worry that there's something really wrong. But the fact is, most problems stem from misunderstood communication. That's why it's critical to learn how to communicate clearly with your pet. I

**Answer 2**
The painting Monalisa was made by Leonardo da Vinci.

**Our Good Judgment without reference (JudgeLM trained with reference drop)**

**Answer 1's Score: 4**          **Answer 2's Score: 6**
Assistant 1's response started off well, providing a detailed description of the Mona Lisa, including the artist, the subject, and the time period. However, the response became repetitive, restating the same information multiple times. Additionally, the last part of the response is completely irrelevant to the question, discussing veterinary and behaviorist advice for pets. This significantly reduces the score for relevance and accuracy. Assistant 2's response, while brief, is accurate and relevant. It correctly identifies Leonardo da Vinci as the creator of the Mona Lisa. However, it lacks detail and does not provide any additional information about the painting, such as the subject or the time period, which would have made the response more helpful. Therefore, Assistant 2's response receives a slightly higher score than Assistant 1, but still not a high score overall due to the lack of detail.

**Our Good Judgment with reference (JudgeLM trained with reference drop)**

**Answer 1's Score: 4**          **Answer 2's Score: 6**
Assistant 1's response started off well, providing detailed information about the Mona Lisa painting, including the artist, the subject, and the time it was painted. However, the response became repetitive, mentioning the same details multiple times. Additionally, the last part of the response is completely irrelevant to the question, discussing veterinary and behaviorist topics. This significantly reduces the score for relevance and accuracy. Assistant 2's response, while accurate and relevant, lacks detail. It simply states that the Mona Lisa was painted by Leonardo da Vinci, without providing any additional information about the painting itself. This makes the response less helpful than it could be. However, it is still more relevant and accurate than Assistant 1's response, hence the higher score.

Figure 14: Good judgment generated by the judge model with reference drop, which addresses the preference for specific fine-tuning formats and gives fair judgments with or without reference.

**Train sample without *swap aug.***

**Input**

**Question**
Create an acronym using the provided letters.
ABCDEF

**Answer 1**
FIDE

**Answer 2**
A Better Comprehension of Digital Education Fund

**GPT-4 Judgement**

**Answer 1's Score: 2**         **Answer 2's Score: 10**
Assistant 1's response is not helpful, relevant, or accurate. The user asked for an acronym using the letters ABCDEF, but Assistant 1 provided an acronym that only uses the letters FIDE, which is not what the user asked for. Therefore, Assistant 1 receives a score of 2.On the other hand, Assistant 2's response is both helpful and accurate. The assistant provided an acronym that uses all the letters provided by the user, which is exactly what the user asked for. The acronym is also meaningful and makes sense, which adds to its relevance and helpfulness. Therefore, Assistant 2 receives a score of 10.

**Train sample with *swap aug.***

**Input**

**Question**
Create an acronym using the provided letters.
ABCDEF

**Answer 1**
A Better Comprehension of Digital Education Fund

**Answer 2**
FIDE

**GPT-4 Judgement**

**Answer 1's Score: 10**         **Answer 2's Score: 2**
Assistant 2's response is not helpful, relevant, or accurate. The user asked for an acronym using the letters ABCDEF, but Assistant 1 provided an acronym that only uses the letters FIDE, which is not what the user asked for. Therefore, Assistant 2 receives a score of 2.On the other hand, Assistant 1's response is both helpful and accurate. The assistant provided an acronym that uses all the letters provided by the user, which is exactly what the user asked for. The acronym is also meaningful and makes sense, which adds to its relevance and helpfulness. Therefore, Assistant 1 receives a score of 10.

Figure 15: An illustration of swap augmentation. We use swap augmentation to exchange the positions of answers, and our GPT-4-generated judgments can be modified correspondingly easily due to their structure.

**Input**

**Question**
There are five birds on a tree, and a person shoots one of the birds. How many birds are left on the tree?

**Reference Answer 1**
One bird was shot and dropped from the tree, leaving four birds remaining on the tree.

**Reference Answer 2**
All the birds have flown away from the tree due to being startled, so there are none left on the tree.

**Answer 1**
There are four birds left on the tree because one bird was shot and fell from the tree.

**Answer 2**
There are no birds left on the tree.

**Our Judgement Output with Reference Answer 1**

**Answer 1's Score: 10          Answer 2's Score: 1**
Assistant 1's answer is accurate, relevant, and helpful. Assistant 1's answer correctly states that there are four birds left on the tree after one is shot, which matches the reference answer. Therefore, it deserves a perfect score of 10. On the other hand, Assistant 2's answer is incorrect and misleading. Assistant 2's answer states that there are no birds left on the tree, which is not true according to the question. Therefore, it receives a score of 1.

**Our Judgement Output with Reference Answer 2**

**Answer 1's Score: 3          Answer 2's Score: 10**
Assistant 1's answer is incorrect. The question states that a person shoots one of the birds, which implies that the bird is killed, not simply startled and falls from the tree. Therefore, there would be four birds left on the tree, not five. However, the reference answer suggests that all the birds would have flown away due to being startled, which contradicts Assistant 1's answer. Therefore, Assistant 1's answer is not accurate or helpful. Assistant 2's answer is accurate, relevant, and helpful. The reference answer suggests that all the birds would have flown away due to being startled. Therefore Assistant 2's answer is correct.

Figure 16: An illustration of changing the reference answer to control model preference. When we change to a different reference answer, the model turns to prefer another answer.

**Input**

**Question**
Given a statement, identify the target audience of the statement.
Take a break from work and enjoy a bowl of ice cream.

**Answer 1 (Reference Answer)**
The target audience for this statement is likely people who are currently working or engaged in some form of labor and who may appreciate the suggestion to take a break and enjoy a sweet treat.

**Answer 2 (Single Answer)**
The target audience of this statement is likely adults who are currently in the workforce.

**Our Judgement Output**

**Answer 1's Score: 10           Answer 2's Score: 7**
Assistant 1 provided a detailed and comprehensive answer, analyzing the statement from different angles and considering the potential needs and desires of the target audience. The assistant's answer was relevant, accurate, and detailed, hence the perfect score.
Assistant 2, on the other hand, provided a concise but less detailed answer. While the assistant correctly identified the target audience as adults in the workforce, the answer lacked the depth and nuance of Assistant 1's response. Therefore, Assistant 2 receives a lower score.

Figure 17: An illustration of grading a single answer.

**Input**

**Question**
Given a statement, identify the target audience of the statement.
Take a break from work and enjoy a bowl of ice cream.

**Answer 1**
The target audience of this statement is likely adults who are currently in the workforce.

**Answer 2**
The target audience of the statement is professional athletes competing in a championship game.

**Answer 3**
This statement is likely directed at those who may be feeling stressed, overwhelmed, or in need of a brief respite from their work tasks. It suggests that they should pause their work for a while and treat themselves to a comforting bowl of ice cream as a way to relax and refresh before returning to their work duties.

**Our Judgement Output**

**Answer 1's Score: 5**          **Answer 2's Score: 1**          **Answer 3's Score: 8**
Assistant 1's answer is not very accurate or detailed. The statement "Take a break from work and enjoy a bowl of ice cream." is not likely to be directed at adults in the workforce specifically. It could be directed at anyone who is working, regardless of their age or occupation. Therefore, the assistant's answer is not very helpful or relevant.
Assistant 2's answer is completely off the mark. The statement does not suggest that it is directed at professional athletes competing in a championship game. This answer is not accurate, relevant, or helpful.
Assistant 3's answer, on the other hand, is quite accurate and detailed. It correctly identifies that the statement is likely directed at people who are feeling stressed or overwhelmed by their work, and suggests that they should take a break and enjoy a bowl of ice cream. This answer is helpful, relevant, and accurate.

Figure 18: An illustration of judging multiple answers.

**Input**

**Question**
Given a statement, identify the target audience of the statement.
Take a break from work and enjoy a bowl of ice cream.

**Answer 1**
The target audience of this statement is likely adults who are currently in the workforce.

**Answer 2**
The target audience of the statement is professional athletes competing in a championship game.

**Our Judgement Output**

**Answer 1's Score: 7**      **Answer 2's Score: 1**
Assistant 1's answer is more relevant and accurate to the question. The statement "Take a break from work and enjoy a bowl of ice cream." is likely targeted at adults who are currently in the workforce. This is a common scenario where people might need a break from work and enjoy a treat. Therefore, Assistant 1's answer is helpful, relevant, and accurate. On the other hand, Assistant 2's answer is not relevant or accurate. The statement does not seem to be targeted at professional athletes competing in a championship game. Therefore, Assistant 2's answer is not helpful, relevant, or accurate.

**JudgeLM Multi-turn Chat**

**Q:** Can you provide a more detailed description of the circumstances in which this statement was made?
**A:** The statement "Take a break from work and enjoy a bowl of ice cream." could be made in a variety of circumstances. Here are a few examples:
  1.A supervisor or manager might say this to their employees during a particularly busy or
  stressful period, encouraging them to take a break and recharge. This would be to ensure the
  employees' well-being and productivity.
  2.A friend or colleague might suggest this during a lunch break or a break in the workday, as a
  way to relax and unwind. This would be a casual, friendly suggestion.
In all these scenarios, the target audience would be adults who are in the workforce. They are the ones who would be likely to relate to the statement and take it as a suggestion or a message.

**Q:** Can you provide an example of a statement where the target audience is teenagers?
**A:** Certainly! Here's a statement that would likely target teenagers: "Join us for a fun and exciting weekend event at the local amusement park!" This statement is likely to appeal to teenagers who are interested in fun activities, socializing with friends, and enjoying a weekend outing. The statement is relevant, accurate, and targets the teenage audience.

**Q:** Can you provide an example of a statement where the target audience is professional athletes competing in a championship game?
**A:** Certainly! Here's a statement that would likely target professional athletes competing in a championship game: "Join us for a special event to celebrate and honor the achievements of professional athletes competing in a championship game!" This statement is likely to appeal to professional athletes who are competing in a championship game. The statement is relevant, accurate, and targets the professional athletes competing in a championship game.

Figure 19: An illustration of multi-turn chat. Users can get more details, advice, examples, etc., by chatting with JudgeLM.

**Input**

**Question**
Given a statement, identify the target audience of the statement.
Take a break from work and enjoy a bowl of ice cream.

**Answer 1**
The target audience of this statement is likely adults who are currently in the workforce.

**Answer 2**
The target audience of the statement is professional athletes competing in a championship game.

**Our Judgement Output**

**Answer 1's Score: 7**              **Answer 2's Score: 1**
Assistant 1's answer is more relevant and accurate to the question. The statement "Take a break from work and enjoy a bowl of ice cream." is likely targeted at adults who are currently in the workforce. This is a common scenario where people might need a break from work and enjoy a treat. Therefore, Assistant 1's answer is helpful, relevant, and accurate. On the other hand, Assistant 2's answer is not relevant or accurate. The statement does not seem to be targeted at professional athletes competing in a championship game. Therefore, Assistant 2's answer is not helpful, relevant, or accurate.

**JudgeLM Multi-turn Chat**

**Q:** Why not give Assistant 1's answer a score of 10?
**A:** 10 would imply that the answer is perfect and there is no room for improvement. While Assistant 1's answer is accurate and relevant, it could have been more detailed by explaining why adults in the workforce are the target audience. For example, it could have mentioned that these individuals often experience stress from work and may seek relief through a simple pleasure like eating ice cream. Therefore, while Assistant 1's answer is good, it is not perfect, hence the score of 7.

**Q:** Can you give a possible statement for the specific group mentioned in Assistant 2's response?
**A:** Certainly, a possible statement for professional athletes competing in a championship game could be: "Recharge with a bowl of ice cream during your break from the intense game." This statement is tailored to the specific needs and circumstances of professional athletes, making it more relevant and helpful to this group.

**Q:** Thanks! I want to get another target audience for the original statement. Can you help me?
**A:** Of course! Another possible target audience for the original statement could be: "Students studying for exams or working on projects." This group might also appreciate a break with a bowl of ice cream, especially if they're feeling stressed or need a quick energy boost. This answer is also relevant and helpful.

Figure 20: An additional illustration of multi-turn chat. Users can get more details, advice, examples, etc., by chatting with JudgeLM.

**Input**

**Question**
==This bike== is a miniaturized version of what full-size model?

**Reference Answer**
harley

**Answer 1**
bicycle

**Answer 2**
==The bike in the image== is a miniropized version of a Harley-Davidson 1200.

**Our Judgment Output**

**Answer 1's Score: 2**       **Answer 2's Score: 8**
Assistant 1's answer is not helpful, relevant, accurate, or detailed. The assistant simply provided a word, "bicycle," which does not match the correct answer, "Harley." This assistant did not consider synonyms or paraphrases, evaluate the correctness of the prediction, or consider the possibility of multiple correct answers. Therefore, Assistant 1 receives a score of 2. Assistant 2's answer is more helpful, relevant, accurate, and detailed. The assistant provided a paraphrase of the correct answer, "Harley-Davidson 1200," which is a valid match. This assistant also considered synonyms, as "Harley-Davidson 1200" is a synonym for "Harley." However, the assistant could have been more precise by directly stating "Harley-Davidson 1200" instead of "the bike in the image is a miniropized version of a Harley-Davidson 1200." Therefore, Assistant 2 receives a score of 8.

Figure 21: An illustration of multimodal judging. Our JudgeLM has the capacity to judge the VQA task without images.

**Input**

**Question**                  **Question Image**
What is located to the right of the shampoo?

**Reference Answer**
conditioner

**Answer 1 (Sampled from Reference Answer Set)**
conditioner

**Answer 2 (Generated by Multimodal Assistant)**
There is a bottle of conditioner located to the right of the shampoo.

**Our Judgement Output**

**Answer 1's Score: 10**       **Answer 2's Score: 10**
Both Assistant 1 and Assistant 2 provided accurate and relevant answers to the question. Assistant 1's answer was concise and directly answered the question, which is why it received a perfect score. Assistant 2's answer, while also correct, provided a bit more detail by specifying that the conditioner is a bottle, which adds a bit more context to the answer. This additional detail is why Assistant 2 also received a perfect score. Both assistants were helpful and precise in their responses

Figure 22: An illustration of multimodal high-score grading on MM-Vet benchmark. The proposed JudgeLM can replace GPT-4 to grade multimodal answers.

---

**Input**

**Question**
What are the appliances in the kitchen in this floorplan?

**Question Image**

**Reference Answer**
oven, and dishwasher

**Answer 1 (Sampled from Reference Answer Set)**
oven, and dishwasher

**Answer 2 (Generated by Multimodal Assistant)**
The appliances in the kitchen in this floorplan include a dishwasher and a refrigerator.

**Our Judgement Output**

**Answer 1's Score: 10**        **Answer 2's Score: 5**
Assistant 2's answer is not accurate because it mentions a refrigerator which is not in the reference answer. Therefore, it gets a score of 5. Assistant 1's answer is accurate and matches the reference answer exactly, so it gets a perfect score of 10

---

Figure 23: An illustration of multimodal mid-score grading on MM-Vet benchmark. The proposed JudgeLM can replace GPT-4 to grade multimodal answers.

---

**Input**

**Question**
What is d in the last equation?

**Question Image**

Solve the following equations:

1) $8x + 11 = 4x + 14$

**Reference Answer**
0.75, or =0.75, or 3/4

2) $7d - 4 = 11d - 9$

**Answer 1 (Sampled from Reference Answer Set)**
3/4

**Answer 2 (Generated by Multimodal Assistant)**
x in the equation is 8x + 11 = 4x + 14.

**Our Judgement Output**

**Answer 1's Score: 10**        **Answer 2's Score: 2**
Assistant 2's answer is incorrect and irrelevant to the question. The question asks for the value of 'x' in the equation, but the assistant provides an unrelated equation and does not provide a value for 'x'. Therefore, it receives a low score of 2. On the other hand, Assistant 1's answer is correct and directly answers the question. The value of 'x' in the equation 0.75, or =0.75, or 3/4 is indeed 3/4. Therefore, it receives a high score of 10.

---

Figure 24: An illustration of multimodal low-score grading on MM-Vet benchmark. The proposed JudgeLM can replace GPT-4 to grade multimodal answers.

