# OpenReview forum: "JudgeLM : Fine-tuned Large Language Models are Scalable Judges"
_ICLR.cc/2024/Conference — Submitted to ICLR 2024_

### Official Review · Reviewer_CPpo · 2023-10-31

**Soundness:** 3 good
**Presentation:** 3 good
**Contribution:** 2 fair
**Rating:** 5
**Confidence:** 4

**Summary:**

The paper proposes a judge model that works in multiple modes like single grading, multiple answer, etc. There is a related previous work PandaLM (under submission in ICLR as well) that have similar content and method. This paper present different sizes of models, and proposes some augmentation on the training data: swap augmentation and reference support/drop. They makes sense. In experiment section, the ablation study results demonstrate the improvement brought by each of the methods. The paper also builds larger datasets to train the judge model.

**Strengths:**

It is good to see that the JudgeLM-33B achieves higher results than its teacher GPT-4 from Table 2 when comparing with human annotation results, which might reveal the necessity of having a dedicated judge model.
The augmentation on the training data methods, i.e. swap augmentation and reference support/drop, do contains novelty to some extent.
Experiments in the paper are well conducted, and good ablation study to give some good insights on problems like effect of model size, augmentation methods, etc.

**Weaknesses:**

While the advantage of first generating scores for answer pairs then generate reasoning makes reasoning generation optional hence could save time, it is not clear whether this is a good strategy. It has been widely known that chain-of-thoughts reasoning can help improve model's reaoning performance. However, this strategy is against the pattern. Though in the experiment results, it doesn't seem to have big issues. It would be good to have comparison on training and testing with the old-fashion chain-of-thoughts style.

How can the paper claim it can judge multimodal model while the model itself cannot take vision data as input? This does not make any sense.

While the paper claims that the conversation capability is retained, it is likely that the training is on the demonstrated question and various model answers presented during training. From the conversation content in Figure 19, the model seems not having conversation on the model's judgement. I don't think this is a good point to emphasize for this paper.

Given there is already a PandaLM paper, while there are still good points and methods proposed in this paper, the novelty is a little weaker.

**Questions:**

It has been widely known that chain-of-thoughts reasoning can help improve model's reaoning performance. However, the paper adopts a different pattern that gives score first then reasoning. Could you explain why this doesn't cause any issue or would the other way works better?

---

> ### Author Response · Authors · 2023-11-22
> **Response to Reviewer CPpo**
>
> Thanks for the valuable comments and suggestions which help us improve this work. Following are our detailed responses to your concerns.
>
> **Question 1:** The model seems not having a conversation on the model's judgment.
>
> **Response 1:** We thank the reviewer for this question. The JudgeLM can have a conversation on its judgments. To prove that, we still take the question in Fig. 19 as an example and list the chat history below.
>
> > Human: Why not give the Assistant 1’s answer a score of 10?
>
> > JudgeLM: 10 would imply that the answer is perfect and there is no room for improvement. While Assistant 1's answer is accurate and relevant, it could have been more detailed by explaining why adults in the workforce are the target audience. For example, it could have been mentioned that these individuals often experience stress from work and may seek relief through a simple pleasure like eating ice cream. Therefore, while Assistant 1's answer is good, it is not perfect, hence the score of 7.
>
> > Human: Can you give a possible statement for the specific group mentioned in Assistant 2's response?
>
> > JudgeLM: Certainly, a possible statement for professional athletes competing in a championship game could be: "Recharge with a bowl of ice cream during your break from the intense game." This statement is tailored to the specific needs and circumstances of professional athletes, making it more relevant and helpful to this group.
>
> > Human: Thanks! I also want to get another target audience for the original statement, can you help me?
>
> > JudgeLM: Of course! Another possible target audience for the original statement could be: “Students studying for exams or working on projects.” This group might also appreciate a break with a bowl of ice cream, especially if they‘re feeling stressed or need a quick energy boost. This answer is also relevant and helpful.
>
>
> **Question 2:** Given there is already a PandaLM paper, while there are still good points and methods proposed in this paper, the novelty is a little weaker.
>
> **Response 2:** Thanks for the question about novelty. Compared with PandaLM, our method has these different contributions:
> - We introduce a high-quality, large-scale dataset for judge models, enriched with diverse seed tasks, LLMs-generated answers, and detailed judgments from GPT-4, laying the foundation for future LLMs evaluating research.
> - We analyze the biases inherent to LLM judge fine-tuning and introduce a series of methods to address them. Our methods significantly improve the consistency of the model in different cases, making the JudgeLM more reliable.
> - We analyze the scaling ability of the language model judge and propose a series of JudgeLM, i.e., JudgeLM-7B, JudgeLM-13B, and JudgeLM-33B, for evaluating LLMs in open-ended scenarios.
> - The proposed judging pattern, i.e., **grading, judging, and reasoning**, makes judging efficient, which only needs little time to grade and judge, and generates time-consuming reasons optionally.
> - The proposed JudgeLM presents extended capabilities as shown in Fig. 1b, including grading single answers, judging multiple answers, judging multimodal models, and multi-turn chat.
>
> **Question 3:** It would be good to have a comparison of training and testing with the old-fashioned chain-of-thoughts style.
>
> **Response 3:** Thanks for your great suggestion. As mentioned in the **Explanation-first Form** part of the global response, we applied the JudgeLM with the explanation-first form, which brings a slight agreement drop and limited consistency improvement.
>
> **Question 4:** How can the paper claim it can judge a multimodal model while the model itself cannot take vision data as input? This does not make any sense.
>
> **Response 4:** Thanks for your question. As mentioned in the **Generalization Ability** part of the global response, the multimodal benchmark, i.e., MM-Vet, proposes to make judgments based on the question text, reference text, and LLM-generated answer using API-based LLM judges. The proposed JudgeLM can also make judgments based on these input texts. Furthermore, we hold the same viewpoint as yours, a better multimodal judge should also `see` the image. So, we plan to use multimodal LLMs, e.g., LLaVA, as the JudgeLM's backbone for better judging multimodal tasks. We leave it as a future work.

---

### Official Review · Reviewer_jzHn · 2023-11-01

**Soundness:** 3 good
**Presentation:** 3 good
**Contribution:** 2 fair
**Rating:** 6
**Confidence:** 4

**Summary:**

The authors proposed JudgeLM, a fine-tuned large language model (LLM) for LLM evaluation. They collected candidate answers with various LLMs and use GPT-4 to generate evaluation, then applied these data to fine-tune Vicuna models. On PandaLM *test* set, JudgeLM achieved better performance than GPT-4, regarding answer correctness.

**Strengths:**

1. The authors used a "grading, judging, and reasoning" form to increase the flexibility of JudgeLM. For example, when only asked to produce a score, JudgeLM cost less time/resource than PandaLM. The authors also argued that such flexibility can help JudgeLM adapt various evaluation scenarios.
2. The authors designed several data augmentation strategies during fine-tuning to help improve the robustness of JudgeLM. Using these methods, JudgeLM was more consistent than GPT-3.5 and PandaLM when chaning candidate answer order or task formulation. They also examined the scaling of JudgeLM, demonstrating that larger models with more fine-tuning data can achieve better performance.

**Weaknesses:**

1. The authors only compared the position bias between JudgeLM and GPT 3.5 or PandaLM, leaving the comparison on the two other biases to within the JudgeLM category. Moreover, They did not report these metrics on GPT-4, the teacher of JudgeLM, to examine if their proposed data augmentation strategies can also help JudgeLM supercede GPT-4 over these aspects.
2. The authors argued that providing external knowledge to LLMs can alleviate the lack of related pre-trained knowledge. However, in their experiments, these "references" are essentially reference answers. Such information might be too sufficient for any LLM evaluation model, greatly decreasing the difficulty of the task, and is usually not available in most real-life scenarios (with only a few exceptions like judging homework submissions). Although incorporating refenrence answers do improve JudgeLM on validation data without reference, it could be better to conduct experiments where knowledge is injected under their retrieved form, instead of a well-organized answer.
3. The authors proposed three data augmentation strategies during fine-tuning. Yet among these methods, "reference drop" is basically equivalent to controlling the amount of samples with reference support, and thus should be a hyper-parameter of the latter strategy instead of an extra method.
4. The authors fine-tuned JudgeLM so that the model provides scores before the explanation. This approach could speed up the evaluation when explanations are neglected. However, it could be better to compare the performance between this evaluation form and the explanation-first one *on JudgeLM itself* instead of with the original PandaLM.

**Questions:**

1. In Table 3, is the total evaluation time of PandaLM computed based on 1 GPU only? If so, why do you not consider such a scenario where multiple (e.g. 8) PandaLMs run parallely on multiple machines with 1 GPU on each machine? This could be more fair to PandaLM, as all models use the same amount of GPU resources (which is the major computation resource).
2. Why is the data in Table 4 not aligned with the ones in Table 1? According to the paper, it seems like the final JudgeLM was fine-tuned on 100k data, which happened to appear in Table 4. Are there other factors that cause the difference, for example the number of training epochs?
3. In section 6.1, Table 1 is referenced as "Table. 1", which is different from other table references.

---

> ### Author Response · Authors · 2023-11-22
> **Response to Reviewer jzHn (1/2)**
>
> Thanks for the valuable comments and suggestions which help us improve this work. Following are our detailed responses to your concerns.
>
> **Clarification:** JudgeLM aims to fine-tune LLMs as scalable judges. In this work, we study the biases in fine-tuning judges and propose methods to alleviate them. The three biases and corresponding metrics are defined as follows:
> 1. position bias: we use consistency to measure the judging results before and after swapping LLM answers, as shown in Table 1 and Table 5.
> 2. knowledge bias: as importing external knowledge helps to improve both objective judging ability and reliability, the agreement (w/ teacher) and consistency (w/ swap) mentioned before are all used to measure this, respectively, as shown in Table 1 and Table 6.
> 3. format bias: this bias indicates that fine-tuned judges perform poorly in different template formats, i.e., w/o reference v.s. w/ reference. So, the metrics in mismatched format are used to measure this, as shown in Table 6.
>
> **Question 1:** The authors only compared the position bias between JudgeLM and GPT 3.5 or PandaLM, leaving the comparison on the two other biases within the JudgeLM category.
>
> **Response 1:** We thank the reviewer for the question about comparison. We further analyze the three biases on PandaLM and apply the proposed methods to it, as shown in Table R8 and Table R9. We first transfer the 3.5K JudgeLM train samples into PandaLM format. Then, we train the PandaLM baseline, i.e., PandaLM*, with LLaMA-7B and the 3.5K train samples in PandaLM format. It can be observed that the PandaLM* also has position bias, which prefers the last answer. Besides, PandaLM* also faces knowledge bias and format bias, as shown in Table R9. Last, we apply the proposed methods to PandaLM*. As shown in Table R8 and Table R9, the proposed methods can also improve the performance of PandaLM*.
>
> | Methods    | Agreement ↑ | Consistency ↑ | Bias ↓ (toward 1st) | Bias ↓ (toward 2nd) | Delta Bias ↓ |
> |------------|:----------------------:|:---------------:|:-------------------:|:-------------------:|:--------------:|
> | PandaLM*   |                 71.30 |         69.59 |              9.43 |             20.98 |       11.55 |
> | + swap aug. |                72.41 |          73.50 |              8.71 |             17.79 |        9.08 |
>
> Table R8. Ablation study of PandaLM for the swap augmentation on our val set.
>
> | Methods  | ft w/ ref? | val w/ ref? | Agreement ↑ | Consistency ↑ | Bias ↓ (toward 1st) | Bias ↓ (toward 2nd) | Delta Bias ↓ |
> |----------|:------------:|:-------------:|:----------------------:|:---------------:|:-------------------:|:-------------------:|:--------------:|
> | *matching format.* |
> | PandaLM* | ❎          | ❎           |                71.32 |         69.59 |              9.43 |             20.98 |       11.55 |
> | PandaLM* | ✅          | ✅           |                75.15 |         74.08 |              8.62 |              17.30 |        8.68 |
> | *mismatched format.*|
> | PandaLM* | ❎          | ✅           |                68.67 |            65.00 |             15.82 |             19.18 |        3.36 |
> | PandaLM* | ✅          | ❎           |                70.04 |         70.56 |              9.71 |             19.73 |       10.02 |
> | *w/ ref. drop.* |
> | PandaLM* | ref. drop   | ❎           |                71.88 |         71.56 |             11.24 |              17.20 |        5.96 |
> | PandaLM* | ref. drop   | ✅           |                75.93 |         74.77 |              9.22 |             16.01 |        6.79 |
>
> Table R9. Ablation study of PandaLM for the reference support and reference drop on our val set.
>
> **Question 2:** Moreover, they did not report these metrics on GPT-4, the teacher of JudgeLM, to examine if their proposed data augmentation strategies can also help JudgeLM supersede GPT-4 over these aspects.
>
> **Response 2:** We thank the reviewer for the suggested comparison with GPT-4. As shown in Table R10, we further list the metrics except for the agreement with GPT-4 itself. JudgeLM-33B achieves higher consistency than GPT-4, which demonstrates that JudgeLM-33B is more robust with position bias.
>
> | Methods     | Agreement ↑ | Consistency ↑ | Bias ↓ (toward 1st) | Bias ↓ (toward 2nd) | Delta Bias ↓ |
> |-------------|:-------------:|:---------------:|:-------------------:|:-------------------:|:--------------:|
> | GPT-4       | -           |         85.82 |               6.10 |               8.10 |            2.00 |
> | JudgeLM-33B |       89.03 |         91.36 |              5.55 |              3.09 |         2.46 |
>
> Table R10. Comparison between GPT-4 teacher and JudgeLM-33B on JudgeLM val set.

---

> ### Author Response · Authors · 2023-11-22
> **Response to Reviewer jzHn (2/2)**
>
> **Question 3:** Although incorporating reference answers does improve JudgeLM on validation data without reference, it could be better to conduct experiments where knowledge is injected under their retrieved form, instead of a well-organized answer.
>
> **Response 3:** Thanks for your great suggestion. As shown in Table R11, we further inject original reference answers into paragraphs with different words, and evaluate JudgeLM-33B with the injected paragraphs as references in a 0-shot setting. It shows that JudgeLM-33B can use references in the retrieved form with a certain loss of agreement and consistency.
>
> | Reference Paragraph | Agreement ↑ | Consistency ↑ | Bias ↓ (toward 1st) | Bias ↓ (toward 2nd) | Delta Bias ↓ |
> |---------------------|:-------------:|:---------------:|:-------------------:|:-------------------:|:--------------:|
> | No                  |       89.32 |         92.37 |              3.62 |              4.01 |         0.39 |
> | 50 words            |       87.78 |         92.35 |              3.00 |              4.65 |         1.65 |
> | 100 words           |       87.69 |         91.84 |              2.62 |              5.54 |         2.92 |
> | 200 words           |       86.77 |         90.48 |              2.70 |              6.82 |         4.12 |
> | 300 words           |       86.25 |         89.26 |              3.13 |              7.61 |         4.48 |
> | 400 words           |       85.59 |         89.00 |              2.98 |              8.02 |         5.04 |
>
> Table R11. Performance of JudgeLM-33B with injected paragraphs as references on JudgeLM val set.
>
> **Question 4:** Yet among these methods, "reference drop" is basically equivalent to controlling the number of samples with reference support, and thus should be a hyper-parameter of the latter strategy instead of an extra method.
>
> **Response 4:** Thanks for your question. We think the reference drop should be an independent method. At first, we argue that judging with or without references are two sub-benchmarks, which require judges to make judgments with internal knowledge or by comparing LLM-generated answers with a reference answer, respectively. The reference drop is not only a simple but effective hyper-parameter, but also an important method that bridges the two sub-benchmarks, which enables the JudgeLM to make judgments in different situations.
>
>
> **Question 5:** Why is the data in Table 4 not aligned with the ones in Table 1?
>
> **Response 5:** Thanks for pointing that out. As shown in `6.2 SCALING ANALYSIS OF JUDGELM`, we analyze the scaling ability of the plain JudgeLM, which does not involve the proposed methods, i.e., swap aug, ref sup, or ref drop. We have underlined this in a revision of our paper.
>
> **Question 6:** In section 6.1, Table 1 is referenced as "Table. 1", which is different from other table references.
>
> **Response 6:** Thanks for pointing out the typo, we have fixed it in a revision of our paper.
>
> **Question 7:** It could be better to compare the performance between this evaluation form and the explanation-first one JudgeLM itself instead of with the original PandaLM.
>
> **Response 7:** Thanks for your advice. As mentioned in the **Explanation-first Form** part of the global response, we applied the JudgeLM with the explanation-first form, which does not bring significant agreement improvement. So, we propose to use a flexible score-first form. Furthermore, the proposed methods, i.e., swap augmentation, reference support, and reference drop, can improve the consistency significantly but bring no more inference costs.
>
> **Question 8:** Using the same computation resources to compare JudgeLM and PandaLM.
>
> **Response 8:** We thank the reviewer for the advice. As mentioned in the **Efficiency Comparison** part of the global response, we made the ablations with only 1 GPU, and the JudgeLM is 16.65 times faster than PandaLM when it does not generate reasons.

---

### Official Review · Reviewer_UdgT · 2023-11-02

**Soundness:** 3 good
**Presentation:** 3 good
**Contribution:** 3 good
**Rating:** 5
**Confidence:** 4

**Summary:**

This paper proposed a way of fine-tuning LLMs to be judges to judge the output of other LLMs.
The construction of the judge fine-tuning data is introduced, the judge is done by first rating the response of LLMs, and then gives a reasoning of the rating, and finally compare two models.
Techniques for avoiding biases in judging LLMs outputs are proposed, and experiments show that the proposed system is a better judge then previously proposed LLM judging systems.

**Strengths:**

1. How to evaluate open-ended generation systems is an important problem, and this work shows a step towards it.
2. The paper is detailed, and provides comprehensive ablations of the proposed system.
3. The proposed augmentations for fine-tuning judge systems could be useful for future researches.

**Weaknesses:**

1. The use of GPT-4 generated judge for the fine-tuning data is the biggest weakness IMO, 1) using GPT-4 generated data could limit the usage of the proposed method (as the GPT-4 license), 2) the main evaluation of the judge system is the agreement with GPT-4, thus training on the GPT-4 generated judges may gives the proposed method a unfair advantage compared to other methods.
2. It could be nice if the paper could include some statistics of the type of question in the evaluation set? Like how many of them are on the topic of code generation or high school math?

**Questions:**

1. I would like to know the author's opinon on weakness 1.
2. Since LLMs are currently still bad at certain reasoning tasks (such as counterfactual[R1]), how can we trust the evaluation results of such a judging system built by LLMs?



[R1] Reasoning or Reciting? Exploring the Capabilities and Limitations of Language Models Through Counterfactual Tasks, 2307.02477

---

> ### Author Response · Authors · 2023-11-22
> **Response to Reviewer UdgT**
>
> Thanks for the valuable comments and suggestions which help us improve this work. Following are our detailed responses to your concerns.
>
> **Question 1:** Using GPT-4 generated data could limit the usage of the proposed method (as the GPT-4 license).
>
> **Response 1:** Thanks for your important concern. We emphasize that the JudgeLM dataset is intended only for academic research and any commercial use is prohibited. Because the OpenAI's terms prohibit developing models that compete with OpenAI, the instruction-tuning datasets generated by the OpenAI's API, i.e., Alpaca, PandaLM, etc., all follow this rule. Thanks for your concerns, we have underlined this in a revision of our paper.
>
> **Question 2:** The main evaluation of the judge system is the agreement with GPT-4, thus training on the GPT-4 generated judges may give the proposed method an unfair advantage compared to other methods.
>
> **Response 2:** Thanks for your question. For a fair comparison, we also evaluate JudgeLM on the PandaLM test set in a zero-shot setting. The PandaLM test set is annotated by humans. As shown in Table 2, the zero-shot results of JudgeLM also outperform other judging methods, i.e., PandaLM, GPT-3.5, GPT-4. Furthermore, JudgeLM also achieves a superior 0-shot judging performance on the multimodal benchmark with human annotation, i.e., MM-Vet, as shown in Table R1.
>
> **Question 3:** It would be nice if the paper could include some statistics on the type of question in the evaluation set.
>
> **Response 3:** Thanks for your advice. We counted the distribution of questions in the JudgeLM val set as shown in Table R6, which has been added to a revision of our paper.
>
> |            |  **culture** | **recommendation** | **finance** | **science** | **technique** | **common-sense** |   **art**   |    **math**   | **private-matter** |  **law**  |
> |------------|:------------:|:------------------:|:-----------:|:-----------:|:-------------:|:----------------:|:-----------:|:-------------:|:------------------:|:---------:|
> | count      |      233     |         482        |     142     |     393     |       42      |        373       |     335     |      250      |         421        |    204    |
> | percentage |     4.66%    |        9.64%       |    2.84%    |    7.86%    |     0.84%     |       7.46%      |    6.70%    |     5.00%     |        8.42%       |   4.08%   |
> |            | **planning** |    **roleplay**    |  **coding** |  **health** |  **writing**  |   **hardware**   | **history** | **geography** |     **others**     | **total** |
> | count      |      309     |         77         |     201     |     278     |      625      |        130       |     243     |      199      |         63         |    5000   |
> | percentage |     6.18%    |        1.54%       |    4.02%    |    5.56%    |     12.50%    |       2.60%      |    4.86%    |     3.98%     |        1.26%       |  100.00%  |
>
> Table R6. Distribution of questions in JudgeLM val set.
>
> **Question 4:** Since LLMs are currently still bad at certain reasoning tasks (such as counterfactual), how can we trust the evaluation results of such a judging system built by LLMs?
>
> **Response 4:** Thanks for your concern. We agree with your concerns about LLM's reasoning performance at certain tasks. Nowadays, NLP researchers are still struggling with proposing LLMs with superior reasoning abilities. JudgeLM also needs the proposed `reference sup` method to enhance the judging ability for out-of-domain or counterfactual tasks, as shown in Fig. 11 and Fig. 16. Notably,  the proposed JudgeLM can benefit from stronger foundation LLMs, e.g., the LLaMA2-7B-Chat-based [1] JudgeLM outperforms the original JudgeLM-7B on all metrics, as shown in Table R7. We hold the same viewpoint as Reviewer rHPM, the research of judge models is critical for the development of LLMs and can benefit from advanced LLMs, establishing a positive cycle.
>
> | Methods              | Agreement | Precision | Recall | F1-score | Consistency |
> |----------------------|:--------:|:---------:|:------:|:--------:|:------------:|
> |*w/o reference*|
> | JudgeLM-7B w/ Vicuna       |   81.11  |   69.67   |  78.39 |   72.21  | 83.57 |
> | JudgeLM-7B w/ LLaMA2-chat  |   83.87  |   73.43   |  80.06 |   75.91  | 85.17 |
> |*w/ reference*|
> | JudgeLM-7B w/ Vicuna       |   84.08  |   75.92   |  82.55 |   78.28  | 84.46 |
> | JudgeLM-7B w/ LLaMA2-chat  |   86.60  |   79.47   |  83.11 |   81.02  | 87.74 |
>
> Table R7. Comparison of different base models for JudgeLM-7B on JudgeLM val set.
>
> [1] Llama 2: Open foundation and fine-tuned chat models

---

### Official Review · Reviewer_rHPM · 2023-11-04

**Soundness:** 2 fair
**Presentation:** 3 good
**Contribution:** 2 fair
**Rating:** 5
**Confidence:** 4

**Summary:**

This work proposes a judge language model (JudgeLM) to evaluate open-end generations of LLMs on various tasks. Judge LM is trained to align his preference with a proprietary teacher LLM, tackling the issue of privacy disclosure of evaluating with a proprietary LLM. To train the JudgeLM, a new and larger dataset is curated to encompass the outputs of 11 open-source LLMs given instructions for various tasks. Furthermore, JudgeLM adopts three training techniques to mitigate the biases of finetuned judge LMs: swap augmentation for position bias, reference support for knowledge bias, and reference drop for format bias. The evaluation of the proposed dataset demonstrates the superior alignment with GPT-4 of JudgeLM compared to open-source baselines and GPT-3.5.

**Strengths:**

1. This work tries to tackle the critical problem of evaluating open-ended answers of LLMs and curate a large-scale dataset with annotation of GPT-4.

2. The proposed judge model is trained with two data augmentation tricks, i.e., answer swap and reference dropping, to be more robust to the answer position and the absence of ground answers.

3. The prompt design enables the extended application of JudgeLM on other tasks, e.g., grading a single answer.

**Weaknesses:**

1. One critical issue of the proposed method is its generalization ability to unseen tasks, which is important for an LM to be a general evaluation toolkit. A smaller LM, fine-tuned to distill the ability of a powerful proprietary LM, may experience a decrease in performance on unseen tasks.

2. The judge model lacks the granularity of its judgments and can only output an overall score. However, there may be many aspects to grade an open answer, e.g., factuality, fluency, novelty, and helpfulness. These fine-grained aspects are not considered in this work.

3. The writings related to efficiency are not very accurate. In “efficiency comparison”, the paper claims that JudgeLM’s superior efficiency is due to PandaLM “not support parallel running”. However, the parallel running is only a trivial implementation detail and should not be counted as a contribution. The main difference is that JudgeLM generates scores first followed by explanations and thus stopping generation before explanations can save time.

**Questions:**

1. Is there any results of the performance in judging multiple answers? The paper has mentioned the format bias resulting from mismatched prompt format. However, the prompt template of judging multiple answers also mismatch that of judging two answers.

---

> ### Author Response · Authors · 2023-11-22
> **Response to Reviewer rHPM**
>
> Thanks for the valuable comments and suggestions which help us improve this work. Following are our detailed responses to your concerns.
>
> **Question 1:** The judge model lacks the granularity of its judgments and can only output an overall score. However, there may be many aspects to grade an open answer, e.g., factuality, fluency, novelty, and helpfulness. These fine-grained aspects should be considered in this work.
>
> **Response 1:** Thank the reviewer for this suggestion. We further ask the JudgeLM to output the fine-grained results through the modified template. We also evaluate the JudgeLM-33B on the JudgeLM val set, as shown in Table R4.
>
> |             | Accuracy | Precision | Recall | F1-score |
> |-------------|:----------:|:-----------:|:--------:|:----------:|
> | factuality  | 89.69    | 77.61     | 84.79  | 80.33    |
> | fluency     | 88.35    | 79.86     | 82.35  | 81.01    |
> | novelty     | 87.98    | 78.63     | 84.85  | 81.12    |
> | helpfulness | 88.16    | 79.30     | 83.88  | 81.26    |
> | all 4 aspects above     | 87.92    | 78.70     | 81.78  | 80.08|
>
> Table R4. JudgeLM fine-grained evaluation results on JudgeLM val set.
>
> **Question 2:** Is there any result of the performance in judging multiple answers?
>
> **Response 2:** We thank the reviewer for the advice. As shown in Table R5, we calculate the consistency results in the judging form of answer pairs and multiple answers. We first generate answers on JudgeLM val set through 3 LLMs, i.e., Vicuna-13B, LLaMA-7B, and alpaca-7B, for evaluation. Then we use pairwise judging and multiple judging to grade answers and rank them, respectively. Last, we compute the consistency between the two ranking results. Please note that `Error Rate@2` indicates the position orderings of two answers are different between the result of paired judgment and the result of multiple judgment, and `Error Rate@3` means the position orderings of three answers are different.
>
> |             | Consistency | Error Rate@2 | Error Rate@3 |
> |-------------|:----------:|:---------:|:---------:|
> | JudgeLM-33B |   93.48 |   6.38 |   0.14 |
>
> Table R5. Performance of JudgeLM-33B in judging multiple answers on JudgeLM val set. We calculate the consistency between the pairwise judging results and multiple judging ones.
>
> **Question 3:** The paper has mentioned the format bias resulting from mismatched prompt format. However, the prompt template of judging multiple answers also mismatches that of judging two answers.
>
> **Response 3:** We thank the reviewer for this question. The judging of multiple answers is not misled by the format bias. We hold the viewpoint that judging multiple answers is an easy extension for JudgeLM, which does not change the basis for judging. As mentioned in `4 Inherent Biases - Format Bias`, format bias means the model judging basis changes from pre-trained knowledge to reference, or vice versa. So, judging in mismatched situations faces format bias, as shown in Table 6, but judging multiple answers does not receive a significant performance drop, as shown in Table R5.
>
> **Question 4:** One critical issue of the proposed method is its generalization ability to unseen tasks, which is important for an LM to be a general evaluation toolkit.
>
> **Response 4:** Thank the reviewer for the question. As mentioned in the **Generalization Ability** part of the global response, we apply JudgeLM to different benchmarks following the 0-shot setting and it achieves superior performance.
>
> **Question 5:** The main difference is that JudgeLM generates scores first followed by explanations and thus stopping generation before explanations can save time.
>
> **Response 5:** Thanks for your question. As mentioned in the **Efficiency Comparison** part of the global response, we launched the ablations with only 1 GPU, and the JudgeLM still has superior efficiency when stops to generate reasons.

---

### Author Response · Authors · 2023-11-22
**Global Response to Reviewers**

We extend our sincere gratitude to both the reviewers and the area chairs for your dedicated time invested in reviewing our paper. We have diligently addressed all of the reviewers' concerns in the corresponding responses, and uploaded a revision of our paper. In summary, there are three common concerns regarding generalization ability (Reviewer rHPM and CPpo), efficiency comparison (Reviewer rHPM and jzHn), and explanation-first form (Reviewer jzHn and CPpo).

**(1) Generalization Ability** (to Reviewer rHPM and CPpo)

To validate the generalization ability of JudgeLM on unseen tasks, we test it on the PandaLM val set as shown in Table 2, and the MM-Vet benchmark as shown in Table R1.

As shown in Table 2, the PandaLM val set is an unseen benchmark for JudgeLM, and is used to validate JudgeLM's generalization ability on the pure-language task. We use the human-annotated results as ground truth, and JudgeLM achieves a superior 0-shot-judging performance.

| Methods              | Accuracy | Precision | Recall | F1-score |
|----------------------|:--------:|:---------:|:------:|:--------:|
| GPT-4 (7-shot)       |   95.58  |   88.63   |  87.79 |   88.04  |
| GPT-4 (0-shot)       |   86.70   |   79.75   |  86.41 |   81.81  |
| GPT-3.5 (7-shot)     |   83.03  |   76.14   |  74.84 |   73.62  |
| JudgeLM-33B (0-shot) |   91.74  |   91.08   |  85.58 |   87.26  |

Table R1. JudgeLM zero-shot evaluation results on MM-Vet.

MM-Vet [1] is a vision-language benchmark for evaluating large multimodal models, which uses GPT-4 or GPT-3.5 as a judge. The API-based judge takes the question text, ground-truth text, and the model's prediction as input, and makes judgments based on them. We first use GPT-4, GPT-3.5, and JudgeLM to judge LLaVA's output, respectively. Then, we collect judgments from human annotators, whose judgments include three situations: completely correct, semi-correct, and completely wrong. Last, we compute the metrics between the LLM judges' judgments and human judgments, as shown in Table R1. It can be observed that JudgeLM outperforms GPT-4 (0-shot) and GPT-3.5 (7-shot). Besides, JudgeLM achieves 2.45% higher precision than GPT-4 (7-shot). Furthermore, JudgeLM can use large multimodal models, e.g., LLaVA [2], as the backbone for better processing the multimodal judging. We leave it as a future work.


**(2) Efficiency Comparison** (to Reviewer rHPM and jzHn)

The root cause of efficiency improvements is our modeling method, i.e., “grading, judging, and reasoning”, and support for parallelization is an engineering optimization. The reason for the parallel judging is that we are committed to making JudgeLM an efficient tool for researchers and developers, benefiting the entire LLM community. Furthermore, we also add an ablation study for JudgeLM without parallel judging to achieve more comprehensive ablations, as shown in Table R2. When generating reasons, JudgeLM's efficiency is similar to PandaLM. Furthermore, JudgeLM is 16.65 times faster than PandaLM when it does not generate reasons.

| Methods | models size | total gpu | generate reason? | total time    |
|---------|:-------------:|:---:|:------------------:|:---------------:|
| PandaLM | 7B          | 1 |✅                | 6 hrs 40 mins |
| JudgeLM | 7B          | 1 |❎                | 24 mins       |
| JudgeLM | 7B          | 1 |✅                | 6 hrs 40 mins |

Table R2. Efficiency comparison for PandaLM and our JudgeLM (without parallel judging) on our val set.

**(3) Explanation-first Form** (to Reviewer jzHn and CPpo)

We further evaluate the performance of JudgeLM-7B with explanation-first (CoT [3]) or score-first (Ours) in Table R3. JudgeLM with CoT performs similar agreement with our score-first baseline but with higher consistency, which means that explanation-first form, i.e., CoT, can alleviate the position bias of fine-tuned judges, but not bring significant agreement improvement. So, we choose the score-first method, which has slightly less consistency but more flexible usage.

| Methods  | Agreement ↑ | Consistency ↑ | Bias ↓ (toward 1st) | Bias ↓ (toward 2nd) | Delta Bias ↓ |
|----------|:-------------:|:---------------:|:-------------------:|:-------------------:|:--------------:|
| score-first (Ours) |       75.87 |         73.45 |             19.83 |              6.72 |        13.11 |
| explanation-first (CoT)     |       75.54 |         74.39 |             15.05 |             10.56 |         4.50 |

Table R3. Performance of JudgeLM-7B with explanation-first (CoT) or score-first (Ours) on JudgeLM val set.

Other concerns are all addressed accordingly in the respective rebuttals. We express our sincere appreciation again to the reviewers and the area chairs for your efforts in reviewing our work.

[1] MM-Vet: Evaluating Large Multimodal Models for Integrated Capabilities

[2] Visual Instruction Tuning

[3] Chain-of-thought prompting elicits reasoning in large language models

---

### Meta-Review · Area_Chair_max8 · 2023-12-06

**Metareview:**

The paper introduces JudgeLM, a fine-tuned large language model for evaluating other LLMs in various generation tasks. A comprehensive dataset, gathering the outputs from 11 open-source LLMs, is used to train JudgeLM. They enhanced the training data with three data augmentation techniques such as swapping, reference support, and reference drop. Experiments reveal that JudgeLM achieves better alignment with GPT-4 compared with baseline models in LLM-based evaluation.

The reviewers recognize the contribution of the paper for efficient and accurate evaluation of LLMs. However, significant concerns remain:

1. Concern on the novelty: Similar approaches, like PandaLM, have previously trained smaller models for LLM evaluation. Thus, the novelty of this work is limited.
2. Concern on the generalizability: How well would the proposed method generalize to other tasks (such as solving math problems or generating code) are not explored.  Furthermore, the current training data does not contain response judgments from aspects of factuality,  fluency, etc., which undermines the generalizability of this work for evaluating LLMs in broader senses.

During the rebuttal, the authors tried to address some of the raised issues by reviewers.  However, the main concerns remain.

In conclusion, while the paper tackles an important problem in LLM evaluation, the limited novelty and generalizability lead to a rejection.

**Justification For Why Not Higher Score:**

The recommendation is primarily based on several key shortcomings of this paper as identified by the reviewers including the concerns on novelty and generalization.

**Justification For Why Not Lower Score:**

N/A

---

### Decision · Program_Chairs · 2024-01-16

Reject